# BIDIRECTIONAL HIERARCHICAL REASONING FOR FINE-GRAINED VISUAL RECOGNITION

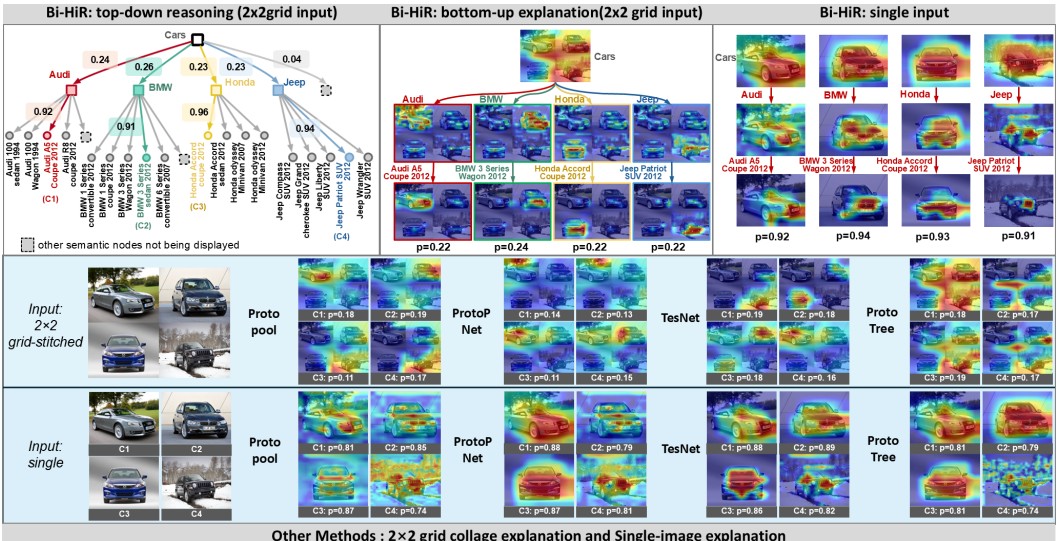

Figure 1: Four images from different categories are concatenated into a 2×2 grid as the input. The results show that: (1) Bi-HiR can discriminate different fine-grained classes and provide trustworthy coarse-to-fine hierarchical explanations at both conceptual and visual level; (2) Other interpretable methods fail to make trustworthy decisions with 2×2 grid input, as well as providing reliable explanations; (3) The visual explanations demonstrate high consistency of Bi-HiR between single input and 2×2 input, while other compared methods fail to provide consistent explanations.

## ABSTRACT

Fine-grained visual recognition (FGVR) requires not only high accuracy but also human-aligned interpretability, particularly in safety-critical applications. While human cognition naturally follows a coarse-to-fine reasoning process—rapid holistic categorization for coarse-grained class followed by attention to local details for fine-grained class—existing post-hoc and ante-hoc interpretability methods fall short in capturing this hierarchy automatically. To address this gap, we propose Bi-HiR, a novel Bidirectional Hierarchical Reasoning framework that emulates human-like cognition by integrating top-down semantic reasoning with bottom-up prototype-based explanations. Specifically, Bi-HiR: **(1)** leverages large language model (LLM)-derived semantic priors to construct coarse-to-fine hierarchies without manual annotations; **(2)** introduces a joint optimization strategy where top-down priors guide bottom-up prototype learning across semantic levels; and **(3)** produces interpretable, step-wise visual and semantic explanations. Experiments on six FGVR benchmarks demonstrate that Bi-HiR achieves competitive SOTA performance and exhibits superior zero-shot generalization. The results also reveal the superiority of Bi-HiR's interpretability on human trust and model error diagnose. Code is publicly available at https://github.com/dudududa-max/Bi-HiR.

## 1 INTRODUCTION

In the era of artificial intelligence, interpretability has become essential, particularly in safety-critical applications such as medical diagnosis, autopilot, and remote sensing, where decisions require transparent and trustworthy evidence Edwards & Veale (2017); Véliz et al. (2021); Mehrabi et al. (2021); Ali et al. (2023). Among these applications, fine-grained visual recognition (FGVR) plays a critical role, yet it remains especially challenging due to its reliance on subtle inter-class differences, which demand precise and semantically rich representations Wei et al. (2021); Ding et al. (2021). Compared to general classification tasks, it not only requires high accuracy but also interpretable reasoning to support human verification Ke et al. (2023); Santra et al. (2022). These challenges underscore the urgent need for models that combine strong performance with explainable decision-making to ensure reliability, accountability, and human trust.

Human cognition in fine-grained recognition follows a coarse-to-fine hierarchy: initial holistic perception rapidly filters coarse categories, followed by knowledge-guided attention shifting to discriminative details. Recent studies have developed post-hoc and ante-hoc methods to enhance interpretability. Post-hoc methods such as Grad-CAM Selvaraju et al. (2017) and LIME Ribeiro et al. (2016) provide explanations after prediction by analyzing feature activations or input perturbations, preserving the original black-box model's performance. However, in fine-grained tasks, their explanations often lack fidelity to the model's true reasoning process and fail to capture the hierarchical progression from coarse to fine semantics. In contrast, ante-hoc methods incorporate interpretability directly into the model via concept layers Bi et al. (2025) or prototypes Sawada & Nakamura (2022), often at the expense of performance. In fine-grained classification, the top-down approaches such as concept bottleneck models (CBMs) Espinosa Zarlenga et al. (2022); Srivastava et al. (2024); Shang et al. (2024); Wang et al. (2024) require costly fine-grained attribute annotations, limiting scalability. While bottom-up methods like part-prototype networks (PPNs) Lin & Wang (2023); Gao et al. (2024); Ma et al. (2023); Wan et al. (2024) learn prototypes automatically but often yield redundant or semantically inconsistent representations, impairing hierarchical reasoning.

This work seeks to retain the high performance of black-box fine-grained recognition models while enhancing their interpretability in alignment with human cognitive processes, thereby promoting generalization through trustworthy, human-aligned reasoning. We consider the black-box model as performing unconscious feature extraction—acting as a "Generalist." On this basis, our method aims to make the subsequent decision-making process transparent, akin to a "Specialist" in FGVR. To this end, we propose a novel **Bi**directional **Hi**erarchical **R**easoning (Bi-HiR) framework by integrating both top-down and bottom-up reasoning at a low cost. The top-down pathway leverages prior knowledge to emulate the human coarse-to-fine cognition process, while the bottom-up pathway identifies discriminative details supporting decisions at each hierarchical node, offering interpretable, step-wise reasoning.

Bi-HiR integrates the strengths of both post-hoc and ante-hoc approaches. It first employs Grad-CAM to localize coarse-grained object regions using the black-box model, highlighting the "Generalist" capability. We then introduce a large language model-driven strategy to automatically construct a coarse-to-fine semantic hierarchy as top-down priors, eliminating the need for manual annotation. Additionally, we propose a bottom-up Bi-HiR optimization framework that learns hierarchical prototypes guided by top-down semantic priors. Experiments on six fine-grained datasets show that Bi-HiR achieves competitive state-of-the-art performance and outperforms both black-box and interpretable models in zero-shot superclass generalization. A user study further confirms its advantage in fostering human trust.

**Contributions. 1)** We propose a novel Bidirectional Hierarchical Reasoning framework for FGVR to emulate human top-down and bottom-up cognition mechanism. **2)** We construct LLM-derived top-down semantic priors and propose a Bi-HiR optimization process, that achieves hierarchical interpretability in both semantic reasoning and visual explanations. **3)** Extensive evaluations demonstrate Bi-HiR achieves competitive SOTA performance and significant improvements zero-shot generalization, human trust, and model diagnose.

## 2 RELATED WORK

**Concept Bottleneck Models (CBMs).** CBMs predict human-understandable concepts before making final task predictions, enabling transparent decision-making and user intervention. The earliest

CBM Koh et al. (2020) relied on explicit manual concept annotations, resulting in high costs and limited scalability. Subsequent studies proposed weakly-supervised methods Schrodi et al. (2024); Liu et al. (2025); Chauhan et al. (2023); Walker (2024); Li et al. (2018) that reduced annotation needs via weakly supervised, unsupervised, or semi-supervised concept discovery and modeling mechanisms. With the rise of multimodal pre-training, recent work Yuksekgonul et al. (2022); Oikarinen et al. (2023); Yan et al. (2023) leveraged vision-language alignment (VLA) to automate concept construction, enabling cross-modal generalization. However, these may prove ineffective in specialized domains where VLA is not readily applicable. Most CBM-based approaches remain constrained by their rigid top-down design requiring predefined fine-grained attributes for concept-driven predictions. In contrast, our approach emulates human-like cognition through coarse-to-fine reasoning by prompting a LLM to act as a fine-grained recognition expert. It first captures holistic coarse semantics, then gradually attends to discriminative details for fine-grained inference without defining the detailed attributes in advance.

**Part-Prototype Networks (PPNs).** Different from CBM methods, PPNs employ bottom-up discriminative part discovery through self-learning visual prototypes. ProtoPNet Chen et al. (2019a) was the first to achieve automatic prototype discovery and visualization without part annotations, but it still has limitations in inference efficiency and interpretability. To improve inference efficiency, numerous methods Rymarczyk et al. (2022; 2021); Kim et al. (2022); Chen et al. (2024); Liu et al. (2023) have reduced the number of prototypes—and thus model complexity—by means such as prototype sharing, compression and merging, and dynamic allocation of prototype numbers. To address interpretability limitations, a substantial body of research Donnelly et al. (2022); Ma et al. (2024); Carmichael et al. (2024); Zhu et al. (2025); Nauta et al. (2021); Wang et al. (2023) has focused on prototype flexibility, spatial localization, and reasoning traceability. While existing PPN-based methods excel at unsupervised part discovery, they often lack semantic coherence between prototypes and concepts and cannot support hierarchical reasoning. This limits their ability to mimic human cognition, which progresses from coarse to fine understanding. In contrast, our approach integrates top-down semantic hierarchy to iteratively refine bottom-up prototypes, enhancing both discrimination and interpretability.

## 3 METHOD

### 3.1 OVERVIEW

Our method transforms a black-box DNN into an interpretable framework, evolving from a "Generalist" to a "Specialist", as shown in Figure 2. We consider that the original CNN has already learned representative features. Building on this, we introduce two modules—Post-hoc Enhancement and Multi-scale Aggregation at "Generalist" stage—to refine pre-trained features and integrate high-resolution information for fine-grained reasoning. At "Specialist" stage, we propose bidirectional hierarchical reasoning (Bi-HiR), combining top-down semantic guidance from a large language model with bottom-up prototype learning. The semantic hierarchy is first constructed via LLM prompting, then Bi-HiR is trained using this hierarchy and fine-grained labels to derive dynamic prototypes for each semantic node. We also illustrate the full Bi-HiR inference process.

### 3.2 POST-HOC ENHANCEMENT

As shown in Figure 2(a), we employ Grad-CAM Selvaraju et al. (2017) to selectively suppress background regions in the input images, thereby enhancing the model's focus on discriminative visual details for fine-grained recognition. Specifically, we generate class activation maps for the top-$k$ predicted categories, normalize and fuse these maps to highlight foreground objects, and then apply a threshold $\tau$ to obtain a binary mask. By masking the original image with the obtained mask, we effectively eliminate background interference, ensuring that subsequent prototype learning focuses on the most discriminative visual regions.

### 3.3 MULTI-SCALE AGGREGATION

To enable Bi-HiR to learn more discriminative features in fine-grained recognition tasks, we propose the multi-scale feature fusion module (Figure 2(b)) to generate multi-scale feature representations $\mathbf{F}_{\mathrm{mf}}$. The detailed implementation is provided in Section A.1.

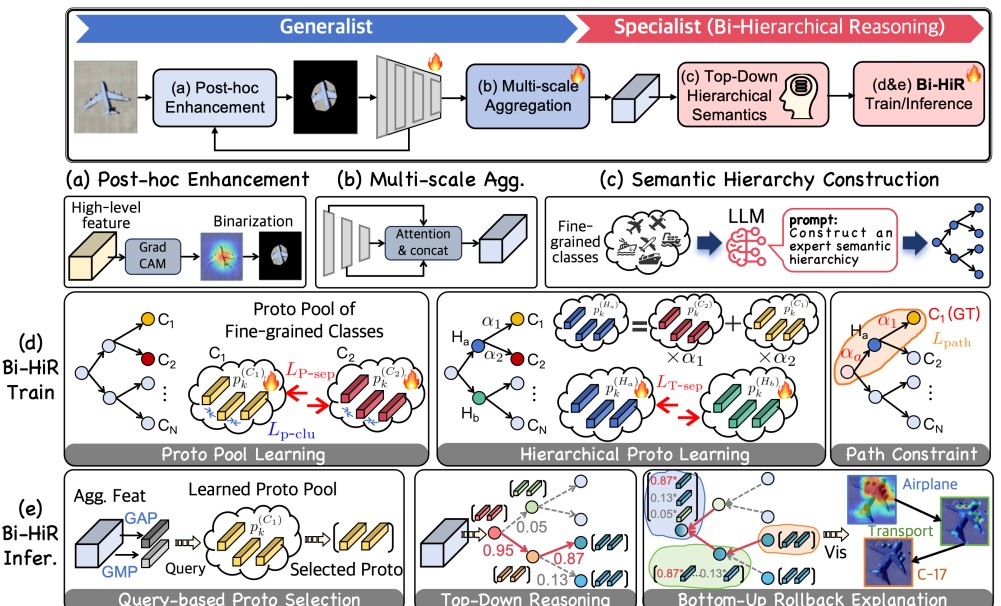

Figure 2: Our proposed architecture consists of five components: (a) Post-hoc Enhancement, (b) Multi-scale Aggregation, (c) Semantic Hierarchy Construction, (d) Bi-HiR Training, and (e) Bi-HiR Inference. Together, these modules enable a top-down, human-like reasoning process that transitions from generalist to specialist.

### 3.4 Semantic Hierarchy Construction

As shown in Figure 2(c), this module constructs a top-down prior for fine-grained recognition by prompting the LLM to act as a domain expert. The prompts include the following steps: **1)** Automatically assign an expert identity based on the fine-grained class domain; **2)** Construct a semantic label hierarchy following the principle of minimal superordinate, integrating multiple perspectives such as visual attributes and functional roles; **3)** Validate and revise the generated hierarchy to ensure logical consistency and scientific accuracy. The detailed prompt texts are given in Section A.2.

### 3.5 Bi-HiR Training

Directly building a semantic hierarchy can degrade model performance despite improving interpretability. To address this, we propose a Bi-HiR optimization method to enhance the performance while maintaining the interpretability in hierarchical semantics and visual explanation. In this way, the visual prototypes for each semantic node are learned guided by the top-down priors within the semantic hierarchy. The details are given as follows.

**Proto Pool Learning.** We posit that each node in the semantic hierarchy requires representative prototypes to establish visual-semantic alignment. To this end, we first initialize a *Proto Pool* for every class using multi-scale aggregated features $\mathbf{F}_{\text{mf}}$ from a pre-trained model, and then optimize it with proposed losses. Specifically, for each training sample in class $N_i$, the global feature vectors via global average and max pooling are obtained to gather the Proto Pool, denoted as $\mathcal{P}_{\text{pool}}^{(N_i)} = \boldsymbol{q}_{\text{GMP}}^{(N_i)} \cup \boldsymbol{q}_{\text{GAP}}^{(N_i)}$. Subsequently, the Proto Pools are jointly optimized using the separation loss ($\mathcal{L}_{\text{P-sep}}$) to increase inter-class distances and the clustering loss ($\mathcal{L}_{\text{P-clu}}$) to improve the compactness of the prototypes within class:

$$\mathcal{L}_{\text{P-sep}} = \frac{1}{N(N-1)} \sum_{i=1}^{N} \sum_{j \neq i}^{N} \max\left(0, m_{\text{sep}} - \|\boldsymbol{\mu}_i - \boldsymbol{\mu}_j\|_2\right) \quad (1)$$

$$\mathcal{L}_{\text{P-clu}} = \frac{1}{N} \sum_{i=1}^{N} \frac{1}{K_i} \sum_{k=1}^{K_i} \left\|\boldsymbol{p}_k^{(i)} - \boldsymbol{\mu}_i\right\|_2^2 \quad (2)$$

where $N$ is the class number, $\boldsymbol{\mu}_i$ and $\boldsymbol{\mu}_j$ are the mean prototype of $\mathcal{P}_{\text{pool}}^{(N_i)}, \mathcal{P}_{\text{pool}}^{(N_j)}$, respectively. $K_i$ is the number of prototypes in $\mathcal{P}_{\text{pool}}^{(N_i)}$, and $\boldsymbol{p}_k^{(N_i)}$ denotes the $k$-th prototype in $\mathcal{P}_{\text{pool}}^{(N_i)}$, $m_{\text{sep}}$ is the margin that enforces a minimum separation between Proto Pools.

**Hierarchical Proto Learning.** Given that neighboring nodes in the semantic hierarchy possess distinct semantic meanings, their corresponding prototypes should maintain clear discriminability. To enforce this constraint, we leverage top-down semantic priors to design an optimization objective for hierarchical prototype learning. First, the internal node $i$ is assigned based on the Proto Pool for its connected nodes $\{j\}_{j=1}^{M_i}$. To start, we select $K$ representative prototypes from Proto Pool for each class for initialization, denoted as $\mathcal{P}_{\text{sel}}^{(N_i)} = \text{Top}_{K/2}\big(\text{sim}(\mathbf{q}_{\text{GAP}}, \mathcal{P}_{\text{sel}}^{(N_i)}\big) \cup \text{Top}_{K/2}\big(\text{sim}(\mathbf{q}_{\text{GMP}}, \mathcal{P}_{\text{sel}}^{(N_i)}\big)$. Let $\mathcal{P}_{\text{sel}}^{(j)} = \mathcal{P}_{\text{sel}}^{(N_i)}$ at first. Then, the prototype sets of the internal node $i$ can be constructed subsequently according to:

$$\mathcal{P}_{\text{sel}}^{(i)} = \sum\nolimits_{j=1}^{M_i} \alpha^{(i \to j)} \mathcal{P}_{\text{sel}}^{(j)} \tag{3}$$

where $\alpha^{(i \to j)}$ denotes the prior probability from the parent node $i$ to its $j$-th child node, and $M_i$ is the number of child nodes connected with node $i$. At the beginning of the training, $\alpha^{(i \to j)}$ is randomly initialized. During training, it is updated as the average probability from the parent node to its child node, aggregated over all batches in the previous training epoch, as shown in the following equation:

$$\alpha^{(i \to j)} = \frac{1}{N} \sum\nolimits_{n=1}^{N} \left( \frac{1}{C_n} \sum\nolimits_{k=1}^{C_n} r_{n,k}^{(i \to j)} \right) \tag{4}$$

where $r_{n,k}^{(i \to j)}$ is the probability that the $k$-th sample in the $n$-th class takes the path from parent node $i$ to its $j$-th child node during the previous training epoch, and $C_n$ is the number of samples in the $n$-th class. Last, we propose a semantic divergence loss for hierarchical prototypes to distinguish different semantic nodes:

$$\mathcal{L}_{\text{T-sep}} = -\frac{1}{Z} \sum_{(a,b) \in \mathcal{I}} \left[ 1 - \text{cosine}\big(\mathcal{P}_{\text{sel}}^{(a)}, \mathcal{P}_{\text{sel}}^{(b)}\big) \right] \tag{5}$$

where $\mathcal{I}$ denotes the set of all internal node pairs in the tree, $Z$ is the total number of node pairs.

**Path Constraint.** To ensure the performance for fine-grained classification and accurate reasoning path, we optimize the prototypes in the Proto Pool using the path loss ($\mathcal{L}_{\text{path}}$) and global classification loss ($\mathcal{L}_{\text{cls}}$). Specifically, at each decision node $i$ (connected with $\{j\}_{j=1}^{M_i}$ nodes) in the hierarchy, the similarity score $S^{(j)}$ between the feature map $\mathbf{F}_{\text{mf}}$ and the prototype set $\mathcal{P}_{\text{sel}}^{(j)}$ is calculated by

$$S^{(j)} = \sum_{x=1}^{H} \sum_{y=1}^{W} \cos\big(\mathbf{F}_{\text{mf}}(x, y), \mathcal{P}_{\text{sel}}^{(j)}\big). \tag{6}$$

where $\mathbf{F}_{\text{mf}}(x, y)$ denotes the feature vector at location $(x, y)$. Then, the next decision node $i + 1$ is decided by selecting the highest similarity score among $\{S^{(j)}\}_{j=1}^{M_i}$, denoted as:

$$i + 1 \leftarrow \arg\max_{j} \{S^{(j)}\}_{j=1}^{M_i} \tag{7}$$

Subsequently, the routing probability $r^{(i \to (i+1))}$ between the decision node $i$ and $i + 1$ can be obtained by:

$$r^{(i \to (i+1))} = \frac{S^{(i+1)}}{\sum_{j=1}^{M_i} S^{(j)}}. \tag{8}$$

To this end, we can obtain the path probability along the decision path as:

$$r_{\text{path}} = \prod_{i=1}^{L-1} r^{(i \to (i+1))}. \tag{9}$$

where $L$ denotes the number of decision nodes in the hierarchy.

Finally, losses $\mathcal{L}_{\text{path}}$ and $\mathcal{L}_{\text{cls}}$ are obtained as follows:

$$\mathcal{L}_{\text{path}} = -\sum_{i=1}^{L-1} \log \left( \frac{\exp S^{(i+1)}}{\sum_{j=1}^{M_i} \exp S^{(j)}} \right) \tag{10}$$

$$\mathcal{L}_{\text{cls}} = -\sum_{i=1}^{N} y_i \log(r_{\text{path}}^{(i)}) \tag{11}$$

where $y_i$ is the ground-truth label of the $i$-th sample, and $r_{\text{path}}^{(i)}$ is the predicted probability of class $i$.

**Loss Function.** The overall optimization of Bi-HiR is to minimize the following loss:

$$\mathcal{L}_{\text{total}} = \mathcal{L}_{\text{P-sep}} + \mathcal{L}_{\text{P-clu}} + \mathcal{L}_{\text{T-sep}} + \mathcal{L}_{\text{path}} + \mathcal{L}_{\text{cls}} \tag{12}$$

## 3.6 Bi-HiR Inference

During inference, the input image undergoes initial processing to extract aggregated features $\mathbf{F}_{\text{mf}}$, which subsequently drive the bidirectional hierarchical reasoning process as follows:

**Query-based Proto Selection.** First, according to the feature map $\mathbf{F}_{\text{mf}}$, top-$K$ prototypes are selected from each class Proto Pool based on the query strategy at training stage. Then, using the $\alpha^{(i \to j)}$ values obtained from the last training epoch as probability weights, the prototypes of internal nodes are constructed according to equation 3.

**Top-Down Reasoning.** After obtaining the prototypes of the internal nodes, the feature map $\mathbf{F}_{\text{mf}}$ of the test sample is compared with the prototypes at each node to compute similarity scores. During inference, the path with the highest probability is selected as the final prediction result. Also, the model calculates the routing probabilities between arbitrary two decision nodes denoted as $\beta^{(i \to j)}$.

**Bottom-Up Rollback Explanation.** During the visual explanation phase, the visual prototypes for each decision node are constructed following equation 3 by replacing $\alpha^{(i \to j)}$ with $\beta^{(i \to j)}$. Next, for each internal node, we compute the similarity between each prototype in the node and every spatial position in $\mathbf{F}_{\text{mf}}$, resulting in $K$ similarity heatmaps. These heatmaps are then upsampled to match the size of the original image. To enhance visual interpretability, the $K$ heatmaps at each node are summed and normalized weighted by $\beta^{(i \to j)}$, and the resulting fused heatmap is superimposed on the original image as the visual evidence for the model's decision.

Table 1: Fine-grained classification performance comparison. Methods are ordered chronologically (2019-2025). The best performance among black-box and interpretable models is marked in blue. Our results, denoted in red with subscripts indicating their global rank among all methods, correspond to the Top-3 performances.

| Method | Year | Venue | Backbone | CUB-200 | Cars | Aircraft | MTARSI | FGSC-23 | FMGT-28 |
|---|---|---|---|---|---|---|---|---|---|
| DCL Chen et al. (2019b) | 2019 | CVPR | ResNet-50 | 87.80 | 94.50 | 93.00 | 95.20 | 89.79 | 90.78 |
| Cross-X Luo et al. (2019) | 2019 | ICCV | ResNet-50 | 87.70 | 94.60 | 92.60 | 91.80 | – | 88.23 |
| ProtoPNet Chen et al. (2019a) | 2019 | NeurIPS | ResNet-50 | 80.20 | 86.10 | – | – | – | 86.42 |
| PMG Du et al. (2020) | 2020 | ECCV | ResNet-50 | 88.20 | 94.20 | 92.40 | 96.20 | 90.95 | 91.57 |
| API-Net Zhuang et al. (2020) | 2020 | AAAI | ResNet-50 | 87.70 | 94.80 | 93.00 | – | – | 91.27 |
| CCBC Du et al. (2021) | 2021 | TPAMI | ResNet-50 | 89.90 | 95.40 | 94.10 | 96.60 | – | 90.97 |
| TesNet Wang et al. (2021) | 2021 | ICCV | Dense-121 | 84.80 | 91.90 | – | – | 88.31 | 89.34 |
| ProtoTree Nauta et al. (2021) | 2021 | CVPR | ResNet-50 | 82.20 | 86.60 | – | 89.21 | – | 86.41 |
| GCP Song et al. (2022a) | 2022 | TPAMI | ResNet-50 | 87.80 | 91.00 | 92.50 | 92.30 | 87.23 | 89.85 |
| TransFG He et al. (2022) | 2022 | AAAI | ViT-B-16 | 91.70 | 94.80 | – | 95.70 | – | – |
| SEB Song et al. (2022b) | 2022 | TPAMI | ResNet-50 | 86.20 | 93.60 | 91.40 | 92.50 | 87.35 | 89.97 |
| ProtoPool Rymarczyk et al. (2022) | 2022 | ECCV | ResNet-34 | 80.30 | 89.30 | – | 90.21 | – | 86.89 |
| ViT-NeT Kim et al. (2022) | 2022 | ICML | DeiT-B | 90.10 | 94.70 | – | 98.47 | 91.23 | – |
| AE-FGM Chang et al. (2023) | 2023 | CVPR | ResNet-50 | 83.60 | 88.60 | 91.30 | 91.20 | 87.12 | 88.35 |
| EFM-Net Yi et al. (2023) | 2023 | TGRS | ResNet-50 | – | – | – | 98.87 | 92.60 | 94.23 |
| ProtoConcepts Ma et al. (2023) | 2023 | NeurIPS | ResNet-34 | 80.70 | 90.40 | – | – | – | – |
| Vmamba Liu et al. (2024) | 2024 | NeurIPS | VMamba-Base | 88.80 | 92.80 | 91.40 | 97.20 | 91.13 | 92.89 |
| Cog-Net Xiong et al. (2024) | 2024 | TGRS | DenseNet-121 | – | – | – | – | 93.02 | – |
| ProtoViT Ma et al. (2024) | 2024 | NeurIPS | DeiT-Tiny | 82.92 | 89.02 | – | 92.17 | 87.65 | 88.43 |
| CGL Bi et al. (2025) | 2025 | TIP | ResNet-50 | 91.80 | 95.97 | 93.60 | 99.20 | 92.89 | 94.63 |
| ProtoSolo Peng et al. (2025) | 2025 | arXiv | ResNet-34 | 74.90 | 89.50 | – | – | – | – |
| **Ours** | 2025 | – | ResNet-50 | 90.63 | 95.19 | 93.27 | **98.99**(3) | 92.65 | **94.32**(3) |
| **Ours** | 2025 | – | ConvNeXt-B | **90.85**(3) | **95.58**(2) | **93.54**(3) | **99.17**(2) | **92.96**(2) | **94.76**(1) |

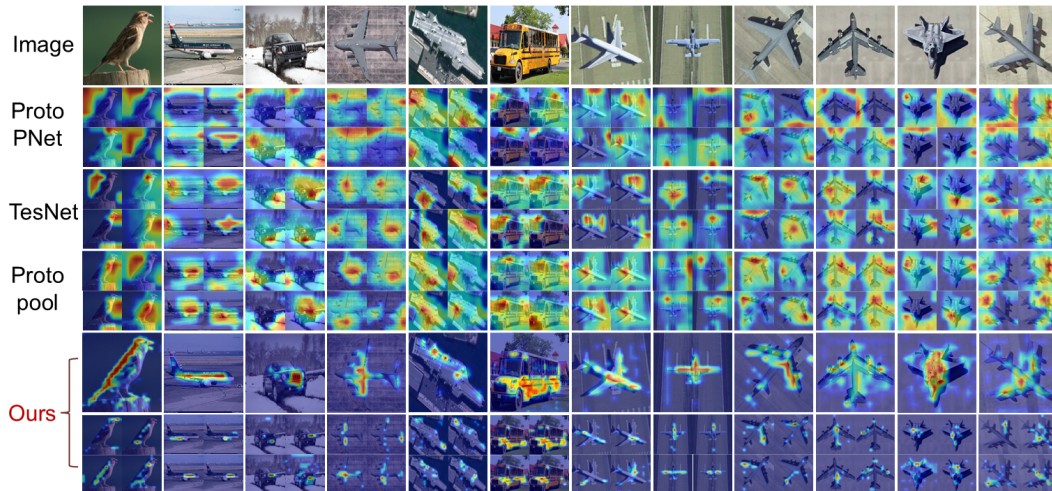

Figure 3: Illustrative comparison of visual interpretability between Bi-HiR and other typical part-prototype networks. The first few rows show the visualization results of category prototypes learned by other methods. The second-to-last row presents the overall visualization of category prototypes learned by our model, while the last row displays the individual visualizations of the category prototypes learned by our model.

## 4 EXPERIMENTAL RESULTS

### 4.1 EXPERIMENTAL SETTINGS

We evaluate the proposed method on six fine-grained classification datasets: CUB-Birds Wah et al. (2011), Stanford Cars Krause et al. (2013), FGVC-Aircraft Maji et al. (2013), MTARSI Wu et al. (2020), FGSC-23 Zhang et al. (2020), and FGMT-28. These datasets encompass a diverse range of categories, including birds, vehicles, aircraft, and remote sensing objects. In the experiments, each prototype in the Proto Pool is of size $1 \times 1792$, with $H, W = 28$. The model is trained for 120 epochs using the Adam optimizer with a learning rate of $1 \times 10^{-4}$. The hyperparameters $k$, threshold, $m_{\text{sep}}$, and $K$ were set to 3, 0.5, 0.2, and 10, respectively. More details of experimental settings are provided in Section A.3.

### 4.2 COMPARISONS WITH STATE-OF-THE-ARTS

Table 1 reports the results of Bi-HiR on six fine-grained benchmark datasets using ResNet-50 He et al. (2016) and ConvNeXt-B Liu et al. (2022) as backbones. Compared with a variety of advanced methods, Bi-HiR consistently delivers competitive performance across most metrics. Figure 3 further presents a visual comparison with representative prototype-based interpretable approaches, showing that Bi-HiR learns prototypes with more distinctive and discriminative target features. More visualization results can be found in Section A.4.

### 4.3 ABLATION EXPERIMENTS

Table 2 evaluates the effectiveness of modules (a)–(e) in Bi-HiR for fine-grained recognition. Row 1 is the black-box baseline. Rows 2 and 3 add, on top of the baseline, (i) a hierarchical structure and (ii) our $\mathcal{L}_{\text{T-sep}}$, respectively. The results indicate that introducing the hierarchy alone leads to a noticeable drop in accuracy, whereas adding the $\mathcal{L}_{\text{T-sep}}$ on top substantially restores performance. Rows 4–6 isolate the contributions of modules (e), (a), and (b). Using (e) or (a) alone yields positive gains, whereas using (b) alone slightly degrades performance. This is because, without (a), unconstrained feature fusion can introduce background cues that are irrelevant to the object. Row 7 shows that jointly applying (a) and (b) suppresses background interference. Rows 8 and 9 compare the model with and without the hierarchy optimized by $\mathcal{L}_{\text{T-sep}}$, and the results show the performance of Row 9 is slightly lower than the best result. Despite the little performance degradation, we claim the

Table 2: Module Ablation. (a) denotes the Grad-CAM method, (b) denotes the multi-scale feature fusion, (c) denotes the hierarchy construction, (d) denotes the $\mathcal{L}_{\text{T-sep}}$, and (e) denotes the Query-based Prototype Selection.

| No. | (a) | (b) | (c) | (d) | (e) | CUB-200 | Cars | Aircrafts | MTARSI | FGSC-23 | FGMT-28 |
|---|---|---|---|---|---|---|---|---|---|---|---|
| 1 | ✗ | ✗ | ✗ | ✗ | ✗ | 89.23 | 93.03 | 91.35 | 96.91 | 91.36 | 92.47 |
| 2 | ✗ | ✗ | ✓ | ✗ | ✗ | 87.12 | 91.24 | 89.21 | 94.95 | 89.22 | 90.93 |
| 3 | ✗ | ✗ | ✓ | ✓ | ✗ | 88.16 | 92.21 | 90.19 | 95.88 | 90.23 | 91.97 |
| 4 | ✗ | ✗ | ✗ | ✗ | ✓ | 91.34 | 95.12 | 93.23 | 98.97 | 92.65 | 94.34 |
| 5 | ✓ | ✗ | ✗ | ✗ | ✗ | 90.65 | 94.57 | 92.66 | 97.22 | 92.74 | 93.89 |
| 6 | ✗ | ✓ | ✗ | ✗ | ✗ | 88.87 | 92.63 | 90.92 | 96.56 | 90.91 | 91.98 |
| 7 | ✓ | ✓ | ✗ | ✗ | ✗ | 90.89 | 94.82 | 92.93 | 97.47 | 92.95 | 94.08 |
| 8 | ✓ | ✓ | ✗ | ✗ | ✓ | **90.97** | **95.71** | **93.68** | **99.29** | **92.92** | **95.04** |
| 9 | ✓ | ✓ | ✓ | ✓ | ✓ | 90.85 | 95.58 | 93.54 | 99.17 | 92.84 | 94.98 |

Table 3: Zero-shot Superclass Generalization, red indicates the highest, blue indicates the second highest. (Abbr: SB=Strategic Bomber, F=Fighter, TR=Transport, CD=Commercial Derivative, CS=Combat ship, AM=Amphibious, SS=Support Ship, C=Civilian ship, PV=Passenger Vehicle, CV=Commercial Vehicle)

| Dataset | MTARSI | | | | FGSC-23 | | | | FMGT-28 | | | | | |
|---|---|---|---|---|---|---|---|---|---|---|---|---|---|---|
| Superclass | SB | F | TR | CD | CS | AM | SS | C | SB | F | PV | CV | CS | SS |
| ResNet-50 | 25.48 | 24.23 | 32.24 | 26.21 | 28.45 | 25.98 | 26.32 | 37.54 | 41.02 | 27.76 | 28.21 | 35.43 | 43.13 | 42.39 |
| DenseNet-121 | 26.89 | 23.98 | 33.09 | 25.96 | 28.78 | 26.43 | 26.87 | 37.58 | 41.45 | 28.29 | 28.76 | 36.01 | 43.64 | 43.02 |
| ConvNeXt_Base | 27.09 | 24.86 | 33.84 | 26.82 | 29.13 | 27.12 | 27.21 | 38.03 | 42.07 | 29.65 | 29.56 | 36.54 | 44.34 | 43.97 |
| ProtoSolo | 31.45 | 32.18 | 36.37 | 32.15 | 34.13 | 33.01 | 33.26 | 42.02 | 42.47 | 29.87 | 32.11 | 41.79 | 45.12 | 45.22 |
| ProtoPNet | 31.89 | 32.61 | 36.94 | 32.67 | 34.58 | 33.43 | 33.68 | 42.35 | 42.85 | 30.13 | 32.58 | 42.19 | 45.46 | 45.76 |
| ProtoTree | 32.02 | 32.68 | 37.09 | 33.06 | 34.74 | 33.69 | 33.84 | 42.58 | 43.01 | 30.27 | 32.76 | 42.38 | 45.79 | 46.01 |
| TesNet | 32.11 | 32.74 | 37.12 | 33.87 | 34.76 | 33.68 | 33.87 | 42.89 | 43.02 | 30.35 | 32.89 | 42.45 | 45.87 | 46.14 |
| ProtoPool | 32.22 | 32.89 | 37.23 | 34.01 | 34.99 | 33.86 | 34.09 | 43.06 | 43.27 | 30.54 | 33.07 | 42.69 | 46.03 | 46.31 |
| ProtoConcepts | 32.31 | 32.99 | 37.32 | 34.12 | 35.12 | 33.97 | 34.27 | 43.21 | 43.38 | 30.72 | 33.25 | 42.82 | 46.17 | 46.43 |
| **Ours** (R50) | 33.23 | 36.45 | 40.26 | 38.42 | 39.54 | 36.75 | 36.85 | 46.87 | 47.73 | 34.81 | 35.85 | 45.67 | 50.76 | 49.43 |
| **Ours** (D121) | 35.43 | 33.67 | 41.25 | 35.63 | 37.97 | 37.03 | 37.32 | 47.35 | 48.13 | 35.21 | 36.32 | 46.31 | 51.21 | 50.96 |
| **Ours** (C_Base) | 36.78 | 37.32 | 42.15 | 39.34 | 39.89 | 38.21 | 38.45 | 49.43 | 48.89 | 36.27 | 38.23 | 47.13 | 52.12 | 51.87 |

superiority of the proposed hierarchy with $\mathcal{L}_{\text{T-sep}}$ on interpretable model diagnose and generalization ability, as demonstrated in Section A.5.4. More ablation studies can be found in Section A.5, including analysis on the top-$k$ and threshold $\tau$ selection in module (a), layer fusion in module (b), hierarchy construction in module (c), and hierarchical prototype discussion in module (e).

## 4.4 ZERO-SHOT SUPERCLASS GENERALIZATION

To assess whether the model learns stable and transferable discriminative features rather than merely memorizing the appearance of leaf-level categories, we design a zero-shot superclass generalization (ZSG) evaluation. The evaluation checks if the inference path reaches the correct superclass on unseen categories, indicating concept-level generalization and hierarchy alignment. Inspired by NBDT Wan et al. (2020), we construct semantically coherent superclasses across six datasets, each grouping multiple fine-grained categories, and sample instances from categories withheld from training, then record whether predictions hit the corresponding superclass node. For black-box baselines, we check whether the predicted class belongs to the target superclass. As shown in Table 3, compared with corresponding black-box models and other interpretable methods, Bi-HiR achieves markedly stronger zero-shot generalization at the superclass level. Detailed settings and additional results are provided in Section A.6.

Table 4: Comparison of subjective interpretability among ante-hoc, post-hoc, and our method based on user study results.

|  | Methods | ACC(↑) | TPR(↑) | TNR(↑) | Efficacy(↑) |
|---|---|---|---|---|---|
| Ante-hoc | ProtoPNet | 52.61 | 44.78 | 60.43 | 7.54 |
|  | ProtoTree | 58.04 | 65.22 | 50.87 | 4.93 |
|  | ProtoPool | 52.61 | 43.04 | 62.17 | 3.77 |
|  | ProtoConcepts | 61.74 | 63.04 | 60.43 | 4.06 |
|  | **Ours** | 86.96 | 86.52 | 87.39 | 79.71 |
| Post-hoc | Integrated Gradients | 52.39 | 50.87 | 53.91 | 1.16 |
|  | Occlusion | 54.35 | 56.52 | 52.17 | 14.49 |
|  | DeepLIFT | 55.65 | 55.65 | 55.65 | 4.35 |
|  | LIME | 53.04 | 42.61 | 63.48 | 3.77 |
|  | Grad-CAM | 61.52 | 65.65 | 57.39 | 2.61 |
|  | **Ours** | 80.65 | 82.17 | 79.13 | 73.62 |

## 4.5 USER STUDY

We also conduct the user study to verify the user acceptance of the provided explanations by Bi-HiR. A questionnaire is designed (details in Section A.7.1) and 115 valid responses are collected. Participants were primarily researchers with deep-learning backgrounds at the Master's and Ph.D. levels, and no significant imbalances were observed in education, machine-learning experience, or vision-related conditions. The questionnaire contained two types of questions. First, given that the model prediction is correct, participants were given explanations from different methods and asked to choose the preferred one. Second, without revealing the predicted label, participants were given the explanations only and asked to judge whether the predictions are correct. More examples can be found in Section A.7.2. We used Accuracy, true positive rate (TPR), and true negative rate (TNR) Labarta et al. (2024) to assess algorithmic effectiveness, and also introduce an Interpretation Preference (IP) metric to capture users' preferences among explanation methods in fine-grained classification (more details about these metrics can be found in Section A.7.3). As summarized in Table 4, Bi-HiR achieves significantly higher Accuracy, TPR, and TNR than competing methods, indicating that it helps users understand model outputs and assess their correctness, increases user trust when predictions are correct, and helps identify reasons for misclassification when predictions are wrong. The IP metric further shows that 73.62% of participants prefer Bi-HiR over other intrinsic methods and 79.71% of them prefer Bi-HiR over post-hoc methods, suggesting a clear subjective preference for explanations generated by Bi-HiR. More analyses of the user study are given in Section A.7.4.

## 5 CONCLUSION

We propose a novel interpretable framework, Bi-HiR. Unlike CBMs and PPNs, Bi-HiR leverages large language models to automatically construct a coarse-to-fine semantic hierarchy without manual annotation. Furthermore, it proposes an optimization strategy where top-down hierarchical priors guide bottom-up prototype learning, thereby enabling a more concise and transparent human-like reasoning process. Extensive experimental results demonstrate that our method achieves classification accuracy comparable to state-of-the-art approaches. Moreover, zero-shot superclass generalization experiments show that Bi-HiR possesses superior generalization capability. In addition, a user study involving 115 participants indicates that the interpretability of Bi-HiR earns higher user trust. Overall, this work provides users with more intuitive, understandable, and trustworthy decision support, and opens up new directions for future research on interpretability.

## 6 ETHICS STATEMENT

This research complies with the ICLR Code of Ethics. We have read and committed to adhering to the relevant guidelines, and we provide clarifications regarding human subjects, data usage, privacy and security, potential risks, fairness, bias, and conflicts of interest. The study does not in-

volve interactions with human participants and does not collect personally identifiable or biometric information. All datasets used are either publicly available or licensed for research purposes and contain no identifiable personal data. The research does not involve system intrusion, data leakage, or the development of methods designed to bypass security controls. All experiments were conducted in secure research environments following standard safety practices. Guided by the principle of promoting social and human well-being, we uphold high standards of scientific excellence and reproducibility, ensuring that experimental settings and limitations are rigorously documented and reported. Furthermore, we adhere to fairness and inclusivity principles, paying attention to and mitigating potential biases in data and evaluation, and explicitly prohibiting any form of discriminatory use or claims.

## 7 REPRODUCIBILITY STATEMENT

To ensure the reproducibility of our results, we provide comprehensive supporting information in the main paper and supplementary materials. All datasets used are either publicly available or licensed for research, with sources and citation details clearly indicated in the paper. The details of model training and experimental setups, including network architectures, hyperparameter configurations, optimization strategies, and evaluation metrics, are thoroughly described in the main text. Additional experimental details are provided in the supplementary materials to allow others to faithfully reproduce our experiments. To further enhance reproducibility, we plan to make the code publicly available after the paper is accepted.

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

## A  APPENDIX

### A.1  MULTI-SCALE AGGREGATION

This Multi-scale Aggregation module is designed based on a multi-scale aggregation strategy, as illustrated in Figure 4. First, feature maps at different scales are individually processed by channel and spatial attention modules to enhance their discriminative capacity, thereby emphasizing salient regions while suppressing irrelevant information. The attention-enhanced multi-scale features are then progressively concatenated, where lower-resolution features are upsampled to align with higher-resolution ones before fusion. Through successive concatenation operations, features from various levels are integrated, which preserves the spatial details of shallow features while incorporating the semantic representations of deeper features. The final aggregated feature thus contains multi-scale and multi-dimensional information, providing a more comprehensive and robust representation for subsequent tasks.

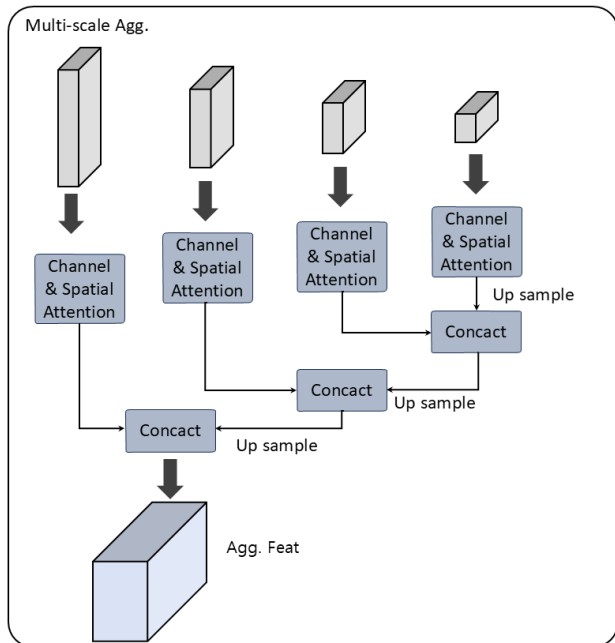

Figure 4: Multi-scale Aggregation.

### A.2  PROMPTS FOR LLM TO CONSTRUCT SEMANTIC HIERARCHY

In this work, we employ GPT-4o, a large language model developed by OpenAI, as the core tool for semantic hierarchy construction. All hierarchical structures are generated through automated interactive prompting with GPT-4o. Specifically, we design generalized, automated expert-level prompts to enable GPT-4o to efficiently construct semantic hierarchies for any dataset, based on the following prompt design principles: 1)Automatic Domain Expert Role Assignment. By automatically parsing the class labels of the input dataset and identifying the task domain (e.g., birds, vehicles, aircraft, etc.), GPT-4o is guided to assume the corresponding expert identity. For instance, for a bird dataset, GPT-4o is set as an ornithology expert with extensive knowledge in biology, taxonomy, and visual recognition; for vehicles or aircraft, it is assigned the relevant domain expertise. This role assignment ensures that the model can fully leverage professional knowledge and background information, enabling it to provide expert-level semantic support for any fine-grained classification task without human intervention. Specific examples are shown in Figure 5.

2)Multi-perspective Hierarchy Organization Principles. During hierarchy construction, GPT-4o is explicitly instructed to follow the "minimal superordinate" principle, grouping similar labels under the same semantic parent node as a priority. The organization process comprehensively considers both visual similarity (e.g., shape, color, texture) and functional attributes (e.g., usage, structural

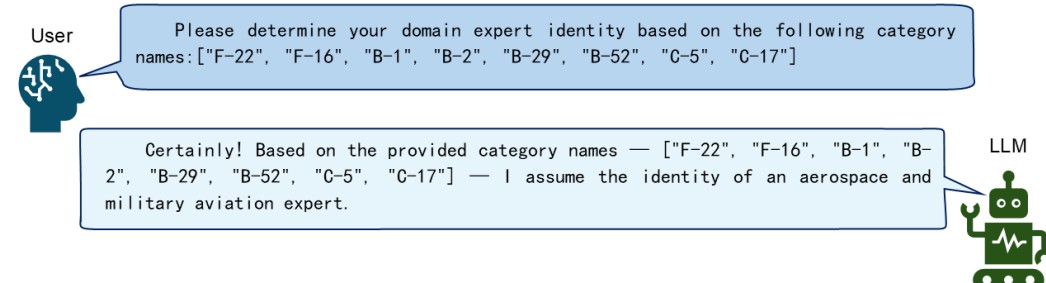

Figure 5: Prompts for domain expert role assignment.

differences) among categories. By automatically discovering and summarizing multi-level semantic relationships, our approach eliminates the need for dataset-specific rules, greatly enhancing generalizability and scalability. Specific examples are shown in Figure 6.

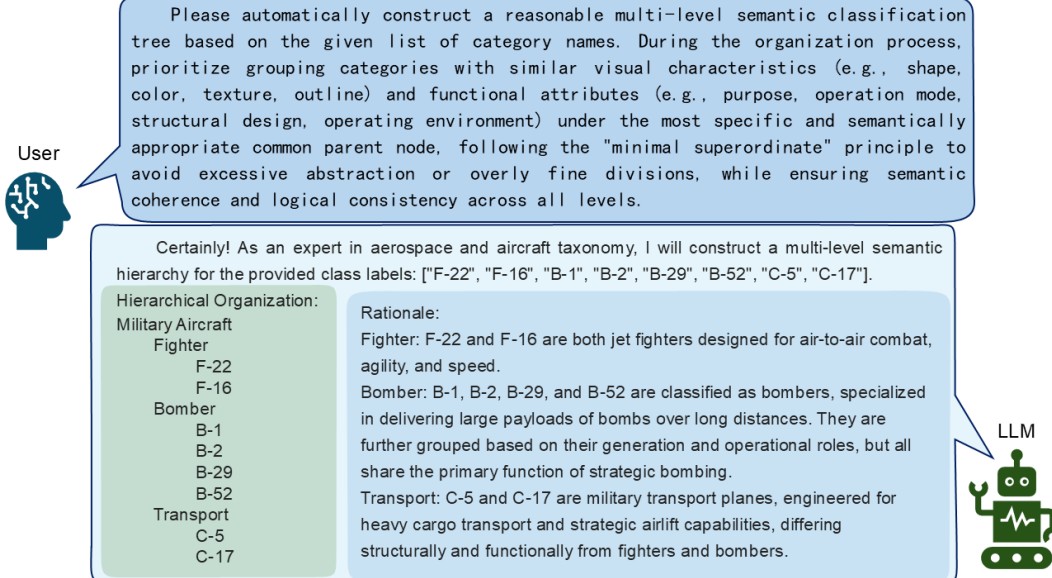

Figure 6: Prompts for hierarchical structure construction.

3)Standardized Tree Structure Output and Automated Validation. GPT-4o outputs the semantic hierarchy in a machine-readable and standardized format (JSON), facilitating downstream model integration and automated processing. Additionally, an automated validation module checks the generated results for format validity, hierarchical consistency, and semantic accuracy. If anomalies are detected, feedback is automatically provided to the LLM for self-correction, minimizing the need for manual intervention. This process can be seamlessly adapted to any new dataset, ensuring robustness and consistency in automated construction. Specific examples are shown in Figure 7.

Based on this, we perform semantic hierarchy construction on six datasets, as illustrated in Figure 8 to Figure 13.

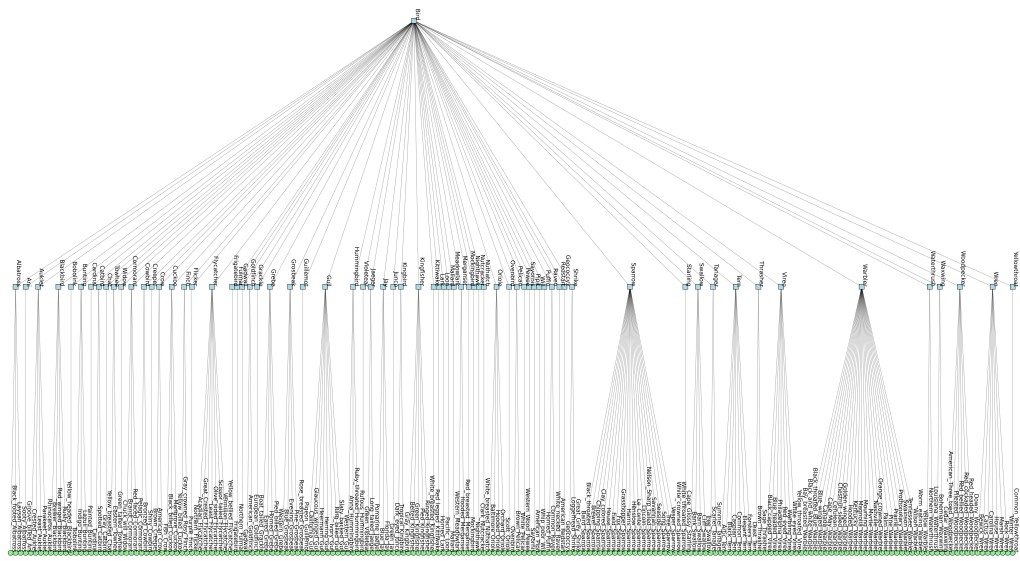

Figure 7: Prompts for hierarchical structure export.

Figure 8: CUB-Birds semantic hierarchy.

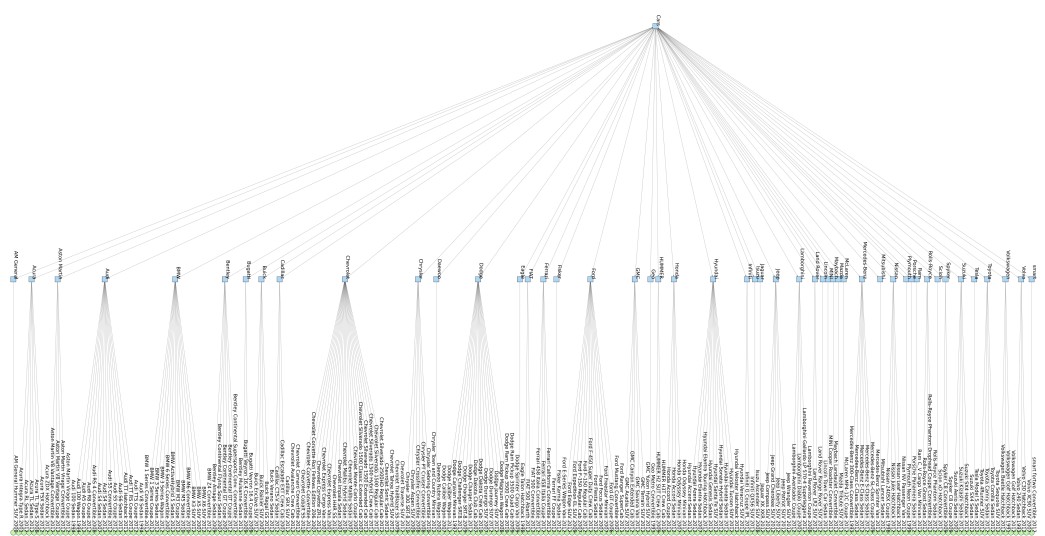

Figure 9: Stanford cars semantic hierarchy.

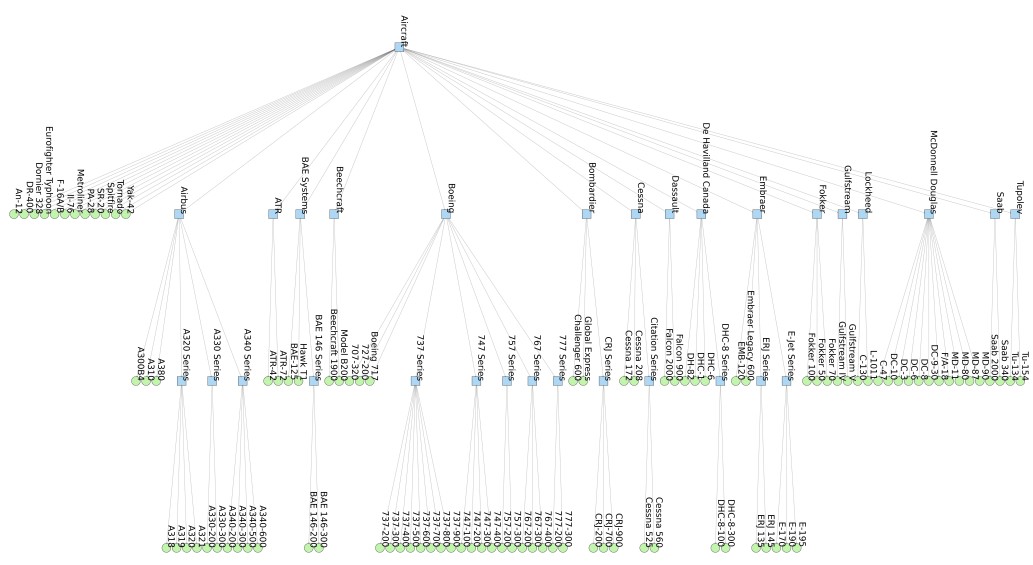

Figure 10: FGVC Aircraft semantic hierarchy.

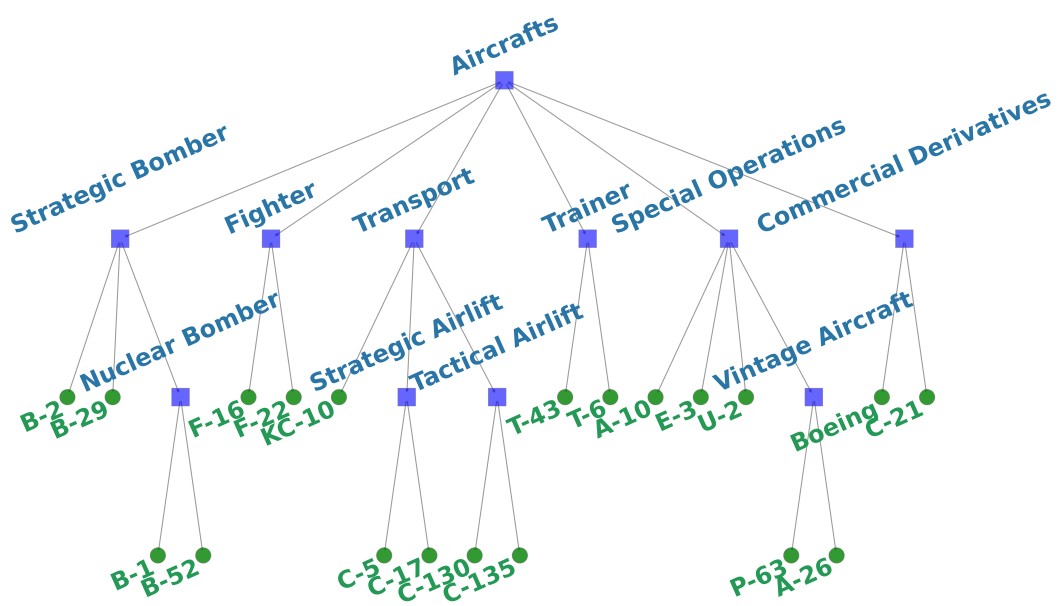

Figure 11: MTARSI semantic hierarchy.

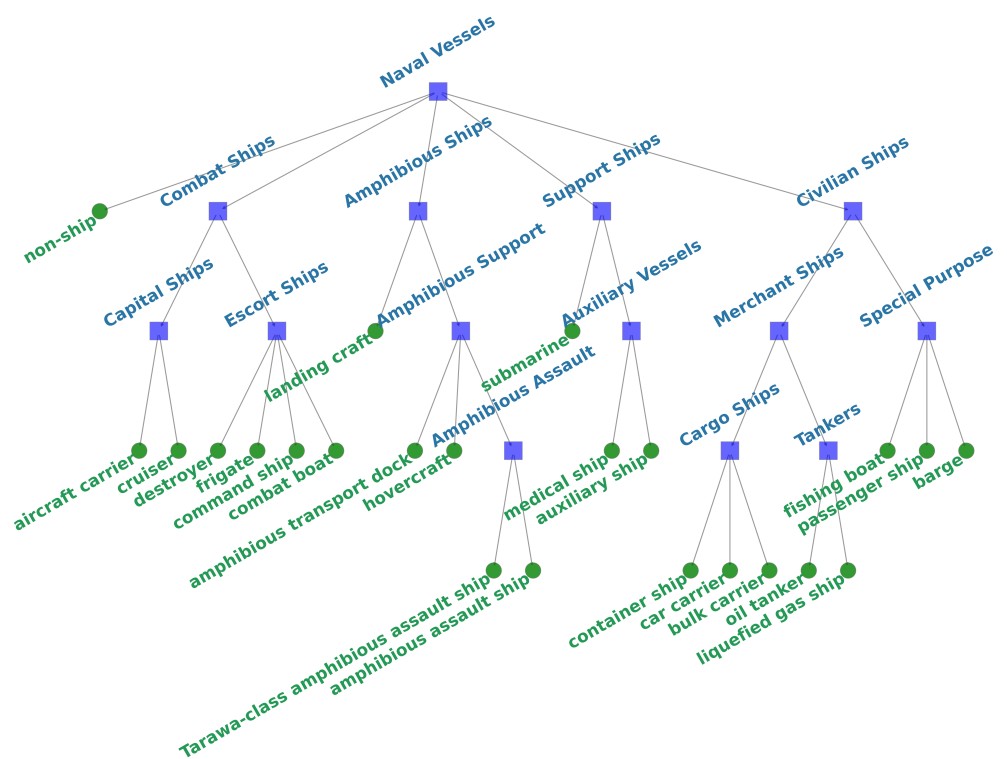

Figure 12: FGSC-23 semantic hierarchy.

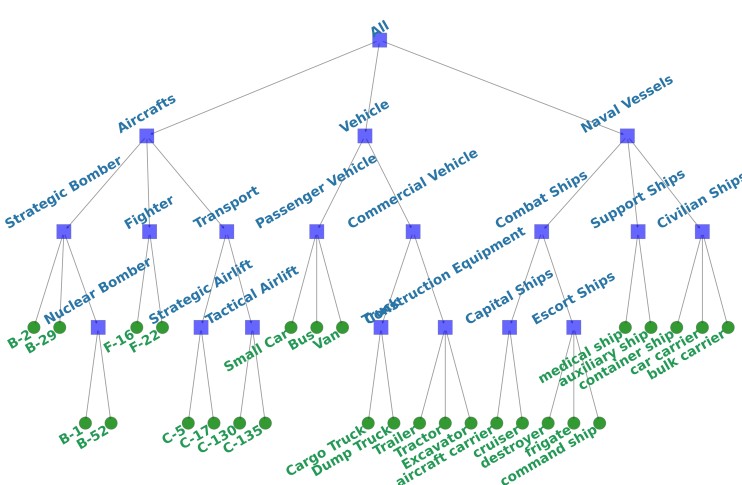

Figure 13: FGMT-28 semantic hierarchy.

## A.3 EXPERIMENTAL SETTINGS

All experiments in this study were conducted in a high-performance computing environment. The hardware configuration included an NVIDIA RTX 3090 GPU, Intel Xeon 6248R CPU (with some experiments run on an Intel Xeon Silver 4316 CPU), and 256GB of memory, with Ubuntu 20.04 as the operating system. The primary programming language was Python 3.9, and the deep learning framework used was PyTorch 2.0. For each experiment, we systematically explored the reasonable ranges of key hyperparameters, such as learning rate, batch size, and the number of prototypes. In the final experiments, we uniformly set the batch size to 32, the number of prototypes to 10, and the learning rate to 1e-4 to ensure the stability and optimal performance of model training. For dataset preprocessing, we applied a series of data augmentation strategies to all input images, including random rotation, horizontal flipping, and cropping, in order to improve the model's generalization ability. All samples were finally resized and cropped to a unified input size of 224 × 224, ensuring consistency across different datasets. To further ensure the reproducibility of experimental results, all random processes involved in the experimental pipeline (such as weight initialization and data shuffling) were controlled by setting fixed random seeds, thus guaranteeing that repeated experiments yield stable and consistent results.

Specifically, CUB-Birds contains 200 classes with 5,994 images for training and 5,794 for testing; Stanford Cars contains 196 classes with 8,144 images for training and 8,041 for testing; FGVC-Aircraft contains 100 classes with 6,667 images for training and 3,333 for testing. In addition, MTARSI comprises 20 aircraft categories with 7,517 images for training and 1,868 for testing; FGSC-23 includes 23 ship categories with 3,264 training images and 816 test images; and FGMT-28 covers 28 categories (vehicles, aircraft, and naval vessels), split into 3,468 for training and 1,024 for testing.

## A.4 VISUALIZATION

In this section, we present more detailed visualization results and further analyze the interpretability of Bi-HiR. For clarity, we simplify CUB-Birds, Stanford Cars, and FGVC-Aircraft to illustrate the decision-making process.

**Visualization of Decision Paths on Six Datasets.** Figure 14 to Figure 19 presents a visualization of the interpretable decision-making process across six datasets constructed in this study. As shown in the figure, we have built a hierarchical structure that incorporates expert reasoning logic to enable interpretable, expert-like judgments. For an input image to be classified, the model first extracts its high-dimensional feature representation through a feature extraction module. In the inference phase, the model starts from the top-level node and sequentially computes the similarity scores between the image features and all child nodes of the current node. At each decision layer, the child node with

the highest similarity score is selected as the next node in the reasoning path, and this process is recursively repeated until a leaf node is reached and classification is completed.

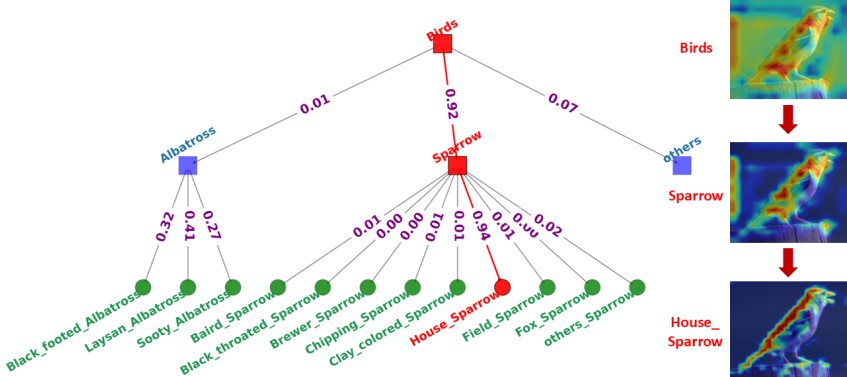

Figure 14: Visualization of Decision Paths on CUB.

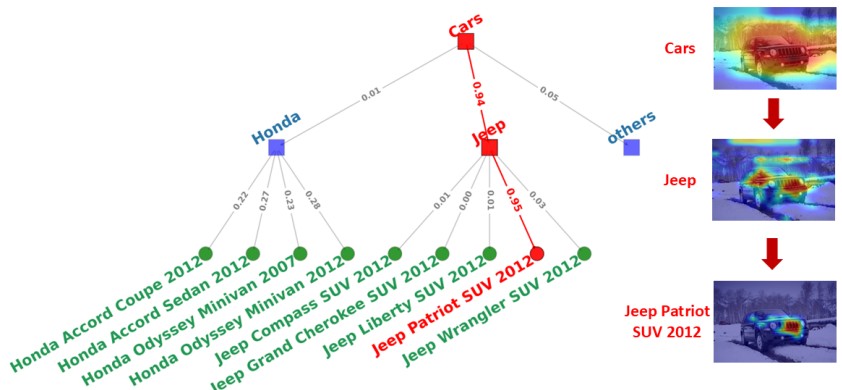

Figure 15: Visualization of Decision Paths on Stanford-Cars.

Specifically, suppose the current inference is at a parent node containing several child nodes (representing classes). The model uses cosine similarity to calculate the similarity between the input image features and each child node's class prototype (as indicated by the edge weights in the figure), selecting the child node with the highest similarity as the next step in the reasoning path. This process clarifies the decision rationale at every layer, making the model's decision logic interpretable and traceable. Moreover, the similarity scores along the decision path provide users with confidence measures for each classification step, which helps in understanding how the final prediction is formed. In cases of misclassification, the inference path can be directly traced back to analyze where the reasoning error occurred, greatly enhancing the model's controllability and transparency in complex real-world scenarios.

Overall, this attribute network structure fully leverages structured semantic priors, transforming the model's end-to-end decision-making process from a "black box" into an interpretable mechanism, thus providing strong support for building highly reliable image recognition systems. Furthermore, we visualize the prototypes corresponding to nodes along the decision path within the original image to indicate the regions the model attends to during each decision, thereby offering visual explanations for the model's reasoning. The visual heatmaps intuitively show the model's focus at different hierarchical nodes. For example, visualizations of prototype activation regions at the Naval Vessels, Combat Ships, Capital Ships, and aircraft carrier nodes reveal the salient structures or key features used for classification at each stage. This not only enhances decision transparency but also enables users to clearly understand how the model gradually converges to the final class. Prototype visualizations are also valuable for troubleshooting and model optimization, as they help identify potential biases along the reasoning path.

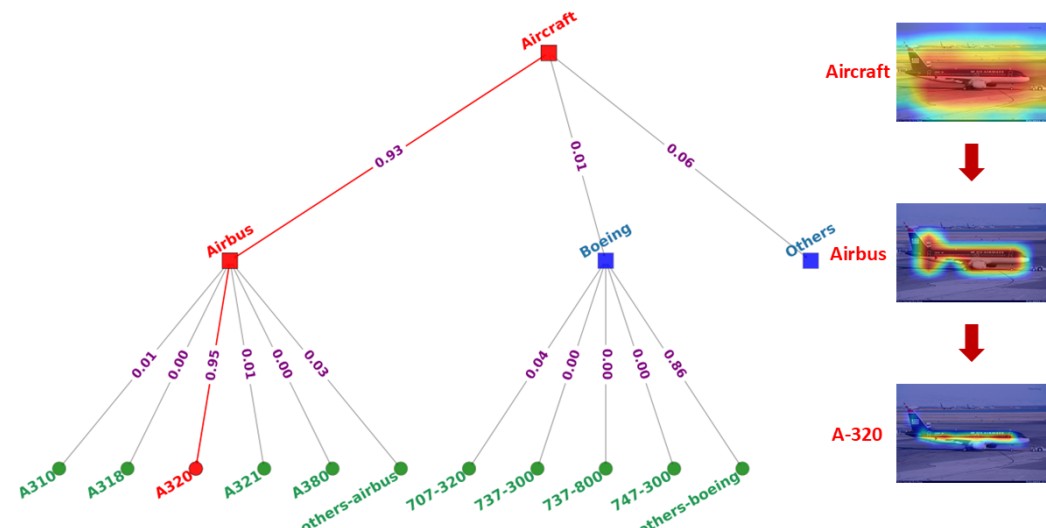

Figure 16: Visualization of Decision Paths on Aircrafts.

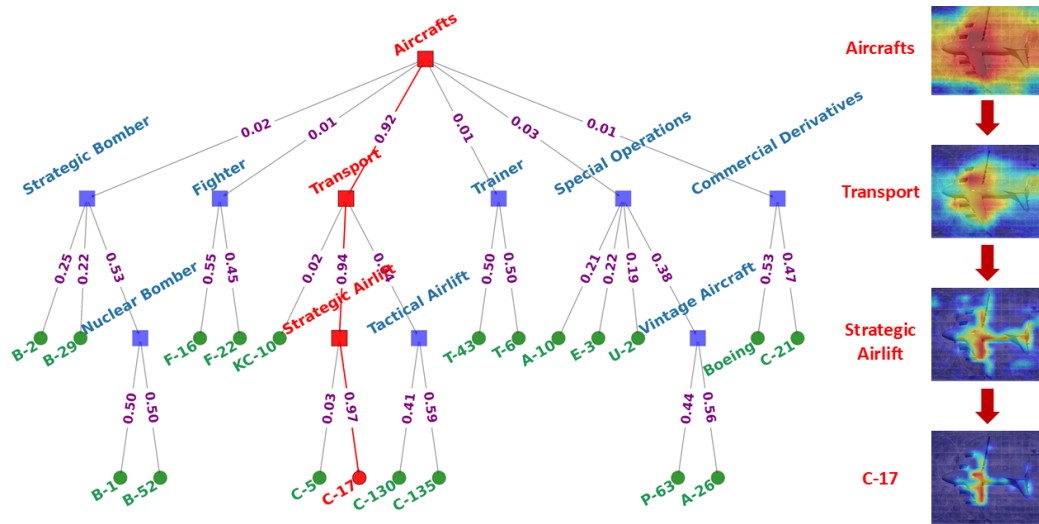

Figure 17: Visualization of Decision Paths on MTARSI.

The specific interpretability process of the model is as follows (using the FGSC-23 dataset as an example): During the testing phase, when an image of an aircraft carrier is input, the model first determines at the "Naval Vessels" node—based on the overall characteristics of the ship—that the target belongs to the "Naval Vessels" category. The highlighted area covers the main structural parts of the hull, displaying typical features of a warship silhouette. Next, at the "Combat Ships" node, the model judges, based on features such as hull layout and superstructure, that the target belongs to the "Combat Ships" category with a probability of 0.93. The salient region corresponds to common deck and cabin structures of support ships. Subsequently, upon entering the "Capital Ships" node, the model further focuses on the distinctive attributes of capital ships and classifies the target as "Capital Ships" with a probability of 0.95. Here, the heatmap highlights are mainly concentrated on typical superstructure and deck details of capital ships. Finally, at the "aircraft carrier" node, the model identifies features such as the aircraft runway and unique equipment layout on the ship, and ultimately classifies the target as "aircraft carrier" with a probability of 0.96. At this stage, the highlighted area accurately covers the unique structures and identifying regions of an aircraft carrier. Throughout the entire decision-making process, the model provides the corresponding probability at each classification node and visually displays the prototype features supporting each judgment as

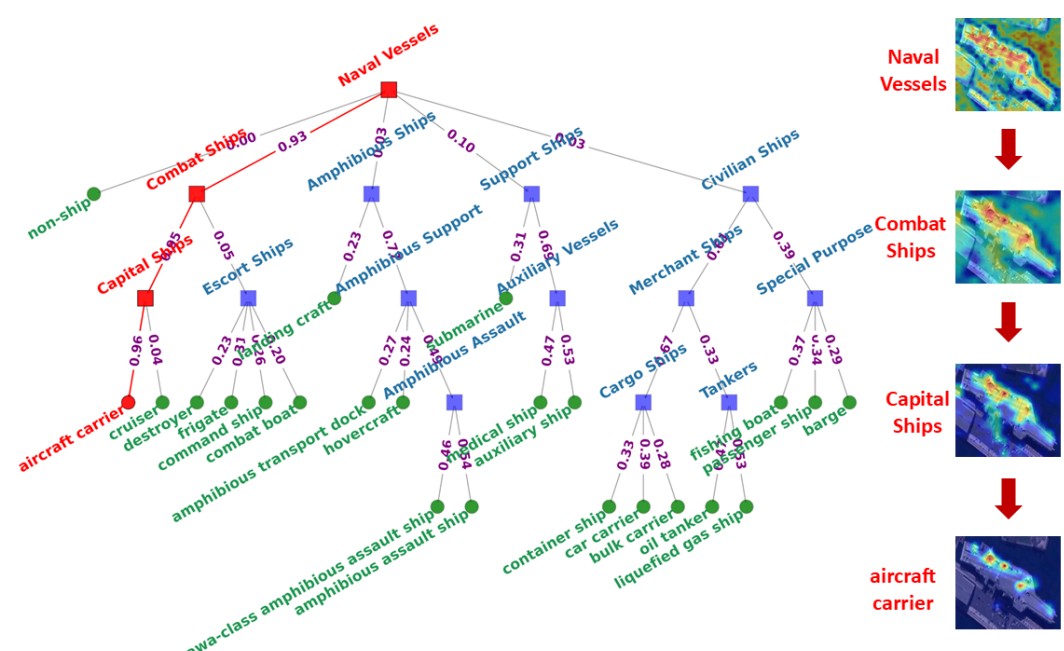

Figure 18: Visualization of Decision Paths on FGSC-23.

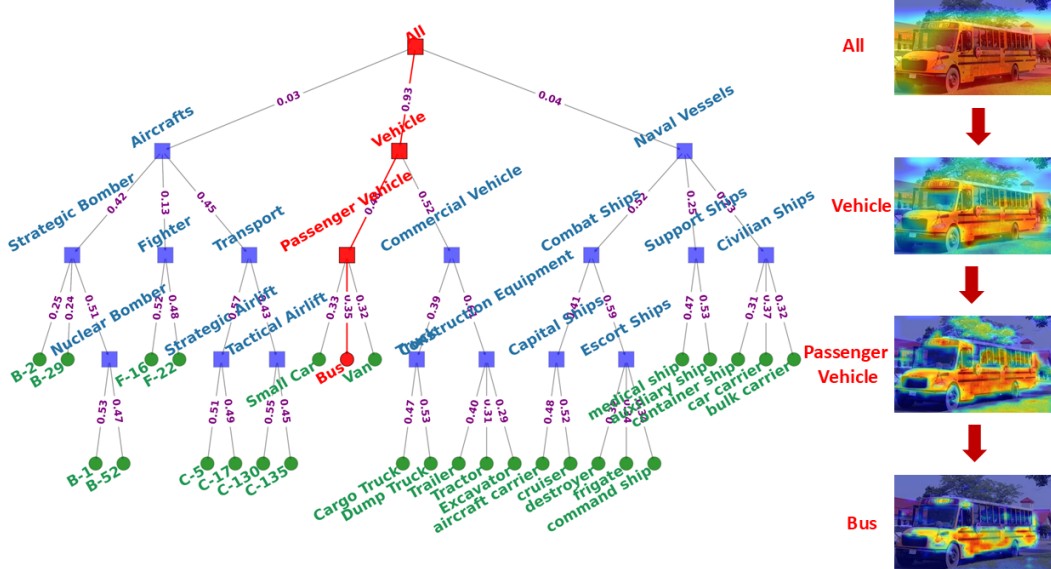

Figure 19: Visualization of Decision Paths on FGMT.

highlighted areas on the original image, thereby realizing an end-to-end, layer-by-layer interpretable decision path. In addition, we further investigated the visualization results of the internal node prototypes, as shown in Figure 21 to Figure 22.

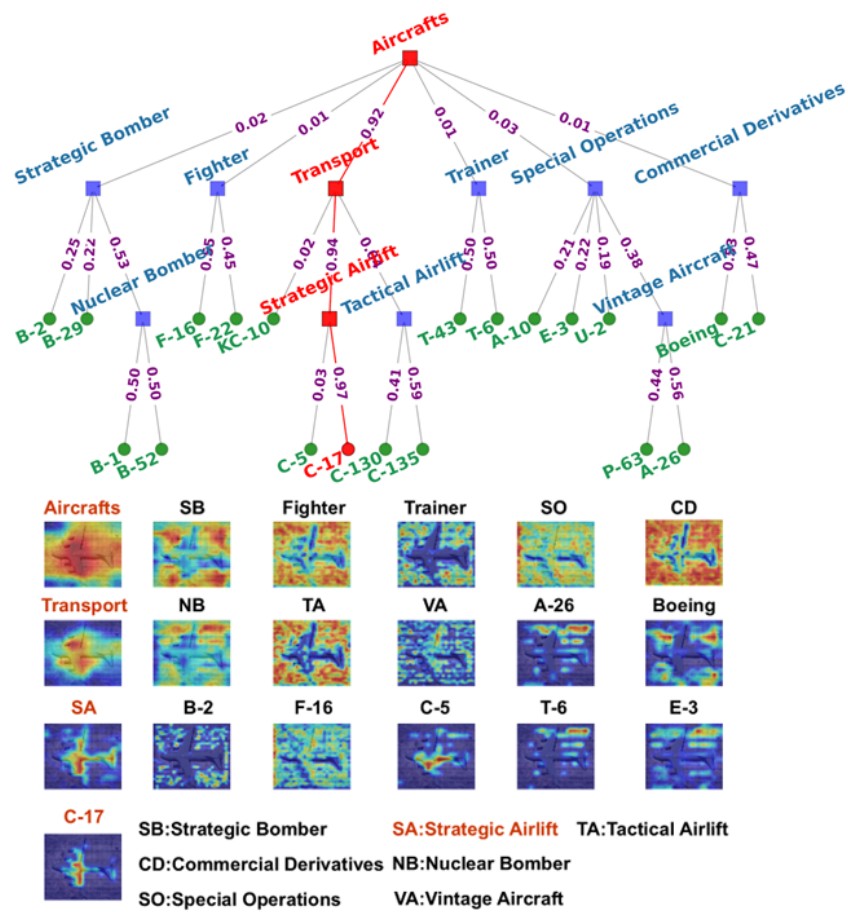

Figure 20: Node-wise Visualizations of Bi-HiR.

## A.5 ABLATION STUDIES

### A.5.1 DISCUSSIONS OF TOP-$k$ SELECTION

As shown in Figure 23, the model attains its best performance when $k$=3, whereas either smaller or larger $k$ values lead to a slight decline. The reason is that when $k$ is too small, the heatmap is dominated by the highest-confidence class, yielding a compact mask that covers only the most salient region of that class and may miss second-best class cues that aid discrimination. As $k$ increases, aggregating heatmaps from multiple classes progressively enlarges the mask, introduces more background unrelated to the target, dilutes attention, increases noise, and thus reduces performance.

### A.5.2 DISCUSSION OF THRESHOLD $\tau$

As shown in Figure 24, the model achieves its best performance when the Threshold is set to 0.5, whereas either smaller or larger Threshold values lead to a slight decline. The reason is that with a too-small Threshold, the resulting mask retains excessive background, making it difficult for the learned visual features to concentrate on discriminative target cues; with an overly large Threshold, the mask preserves only a small portion of the foreground, preventing the model from sufficiently learning the target's discriminative features.

### A.5.3 DISCUSSION OF MULTI-SCALE AGGREGATION

As shown in Table 5 and Figure 25, the fusion location across feature layers affects both model performance and interpretability. We compare three candidate target layers—Layer1, Layer2, and

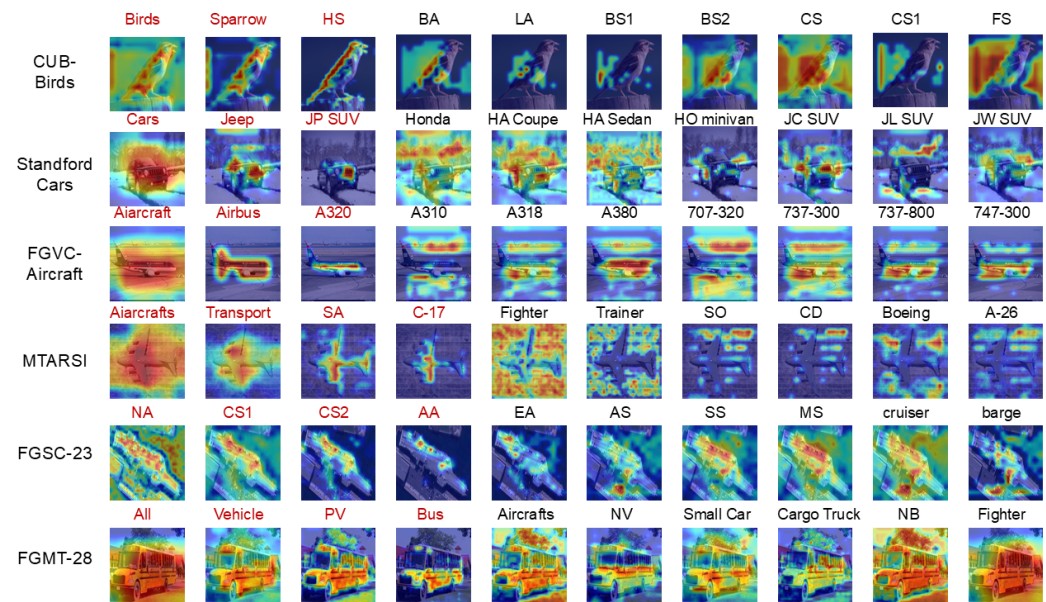

Figure 21: Node-wise Visualizations of Bi-HiR. (Abbr: HS=House Sparrow, BA=Black footed Albatross, LA=Laysan Albatross, BS1=Baird Sparrow, BS2=Brewer Sparrow, CS=Clay colored Sparrow, FS=Fox Sparrow, JP=Jeep Patriot, HA=honda Accord, HO=Honda Odyssey, JC=Jeep compass, JL=Jeep Liberty, JW=Jeep Wrangler, SA=Strategic Airlift, SO=Specoal Operations, CD=Commercial Derivatives, NA=Naval Vessels, CS1=Combat Ships, CS2=Capital Ships, AA=aircraft carrier, EA= Escort Ships, AS=Amphibious Ships, SS=Support Ships, MS=Merchant Ships, PV=Passenger Vehicle, NV=Naval Vessels, NB=Nuclear Bomber)

Layer3. Placing the fusion at Layer2 yields the best accuracy and interpretability; therefore, we adopt Layer2 as the target fusion layer in our model.

Table 5: Performance comparison of multi-scale aggregation on various datasets

| No. | CUB-200 | Cars | Aircrafts | MTARSI | FGSCR-23 | FGMT-28 |
|-----|---------|-------|-----------|--------|----------|---------|
| 1 | 90.83 | 95.47 | 93.54 | 99.04 | 92.76 | 94.76 |
| 2 | **90.85** | **95.58** | **93.54** | **99.17** | **92.84** | **94.98** |
| 3 | 89.93 | 94.47 | 92.74 | 98.63 | 91.97 | 93.49 |

### A.5.4 DISCUSSION OF HIERARCHY CONSTRUCTION WITH OPTIMIZATION

We noticed in Table 2 that introducing hierarchy would degrade the model performance slightly, although the proposed optimization would relieve the performance degradation. However, we claim that this slight degradation is acceptable as the proposed hierarchy construction would improve the interpretability that contributes to model diagnose and better generalization. To verify this, two models with and without hierarchy construction are trained, denoted as M/w and M/wo, respectively.

First, we conduct an error analysis as given in Figure 26. A picture of "C-17" is input to M/w and M/wo to obtain the prediction. It can be found that both models output wrong predictions of "C-5". The explanation provided by M/wo highlights the wings, but we are not able to diagnose the wrong prediction according to the explanation. As a comparison, M/w provides clear decision path and illustrates that the wrong prediction happens at the decision node of discriminating "C-17" and "C-5". The explanations are more trustworthy and the users can diagnose the model more easily, like "the model would be confused about the fine-grained features of C-17 and C-5".

Second, we also conduct a generalization test in Figure 27. An image of "F-35" that is not in the training dataset (but we know that "F-35" belongs to "Fighters") is inputted to M/w and M/wo to

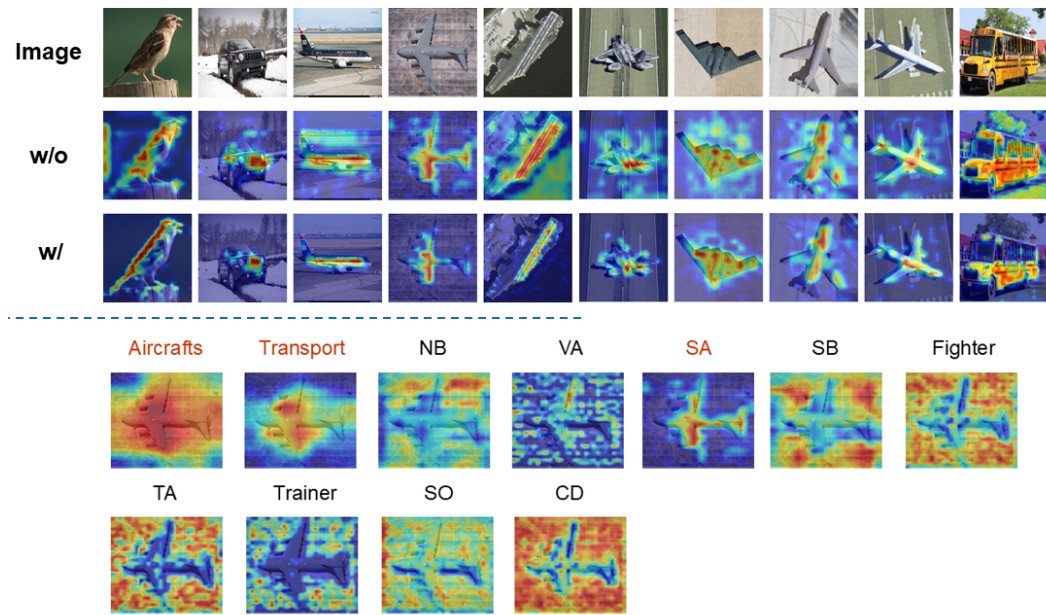

Figure 22: Impact of $\mathcal{L}_{\text{T-sep}}$ on Visual Prototypes, The upper part shows the visualization of leaf node prototypes, while the lower part displays the visualization of internal node prototypes. (Abbr:SB=Strategic Bomber, SA=Strategic Airlift, TA=Tactical Airlift, CD=Commercial Derivatives, NB=Nuclear Bomber, SO=Special Operations, VA=Vintage Aircraft)

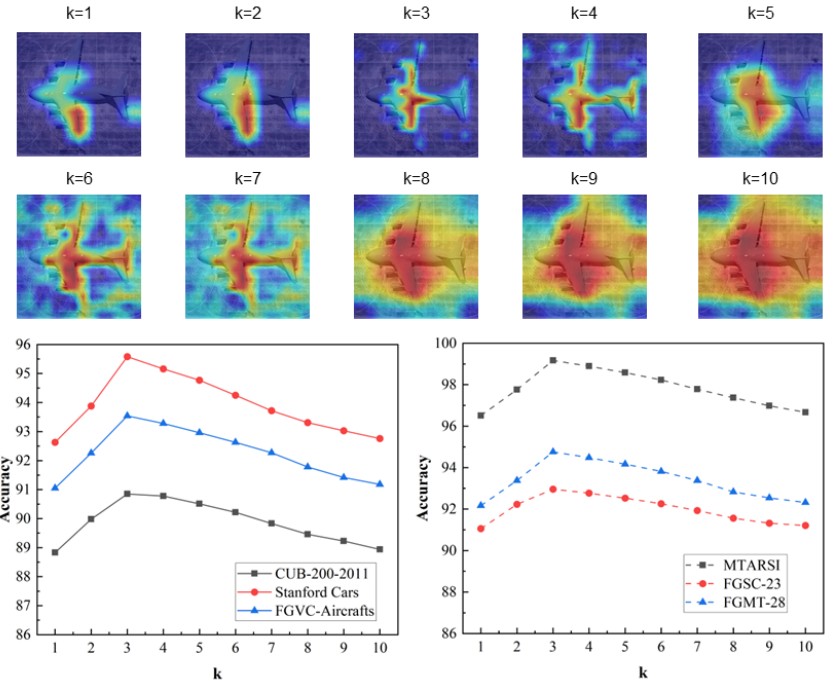

Figure 23: Impact of k

obtain the prediction. The results show that M/wo classifies it as B-52 while M/w classifies it as F-16. Clearly, the prediction of M/w is more trustworthy since the explanations demonstrate it reaches correct super-class node of "Fighter" with good explanations and only confuses to classify the more

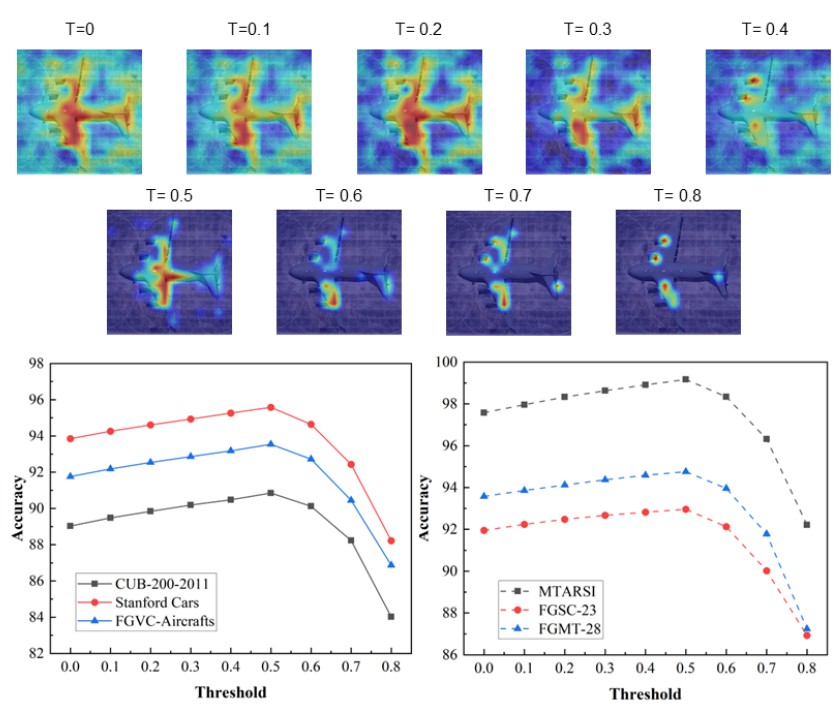

Figure 24: Impact of Threshold

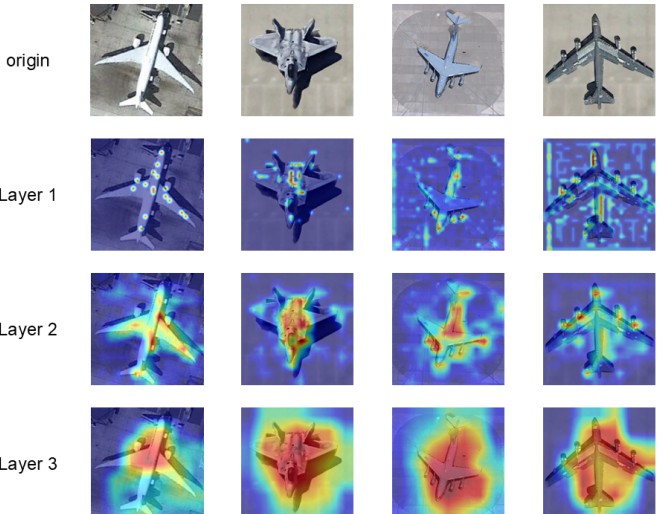

Figure 25: Impact of Multi-scale Aggregation

fine-grained category. As a comparison, M/wo predicts it as B-2 and has no idea whether it is similar with other "Fighter" categories.

### A.5.5 DISCUSSION OF HIERARCHICAL PROTOTYPES

As shown in Table 6 and Figure 28, different strategies for constructing internal-node prototypes affect both model performance and interpretability. We compare three strategies: S&N (sum-then-normalize), Avg (weighted averaging), and P-w (aggregation weighted by path probabilities). Among them, P-w delivers the best results in terms of accuracy and explanatory consistency, with S&N and Avg performing slightly worse. These findings suggest that incorporating path prob-

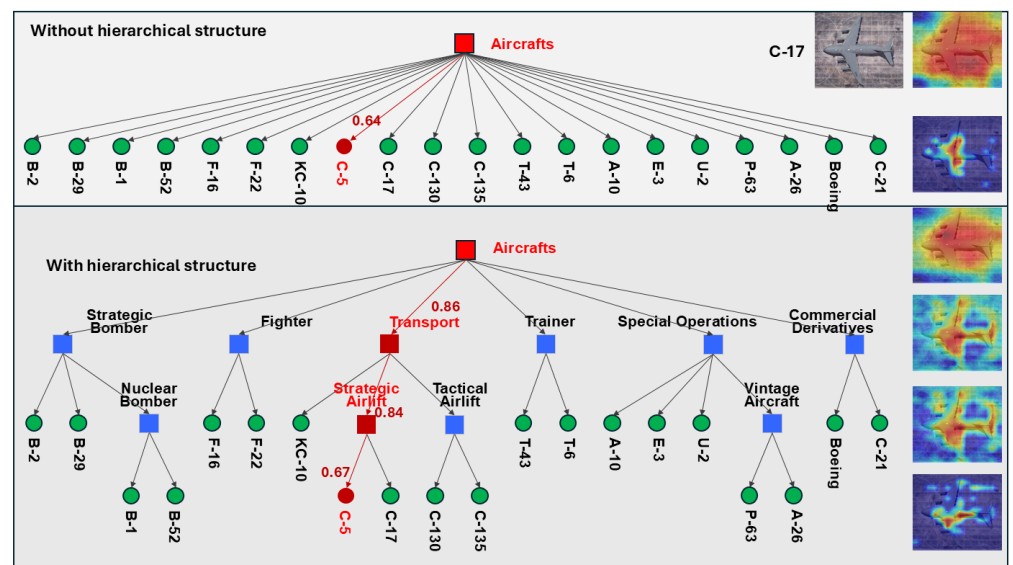

Figure 26: The impact of (c) and (d) on prediction errors.

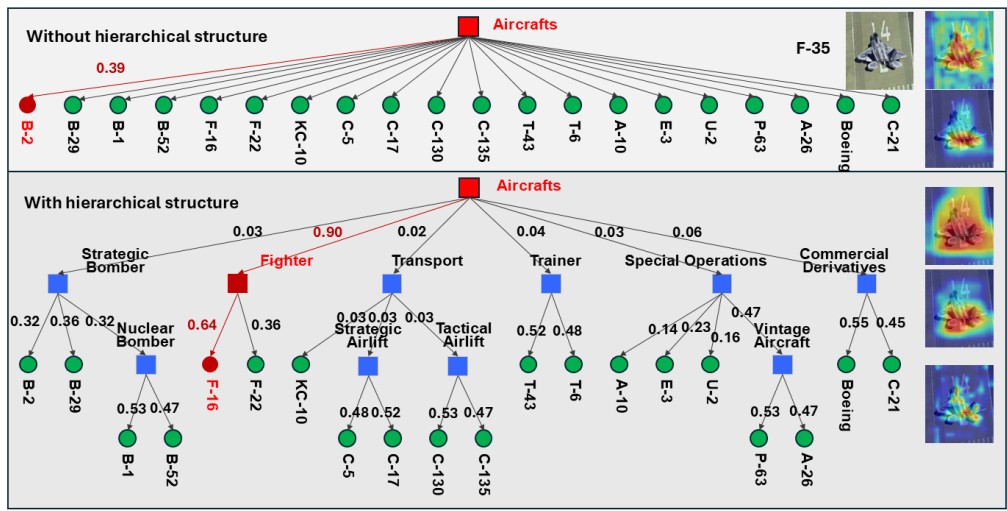

Figure 27: The impact of (c) and (d) on superclass predictions.

abilities to encode node importance in hierarchical reasoning improves predictive reliability and strengthens semantic alignment.

Table 6: Performance comparison of hierarchical prototype on various datasets

|       | CUB-200 | Cars  | Aircrafts | MTARSI | FGSCR-23 | FGMT-28 |
|-------|---------|-------|-----------|--------|----------|---------|
| S&N   | 90.61   | 95.37 | 93.38     | 99.01  | 92.65    | 94.63   |
| Avg   | 90.67   | 95.42 | 93.41     | 99.08  | 92.73    | 94.79   |
| P-w   | **90.85** | **95.58** | **93.54** | **99.17** | **92.84** | **94.98** |

### A.5.6 DISCUSSION OF PROTOTYPE NUMBER

As shown in Figure 29, model performance increases progressively as the number of prototypes $K$ grows; it stabilizes at $K$=10, and further increases in $K$ lead to performance saturation. Although

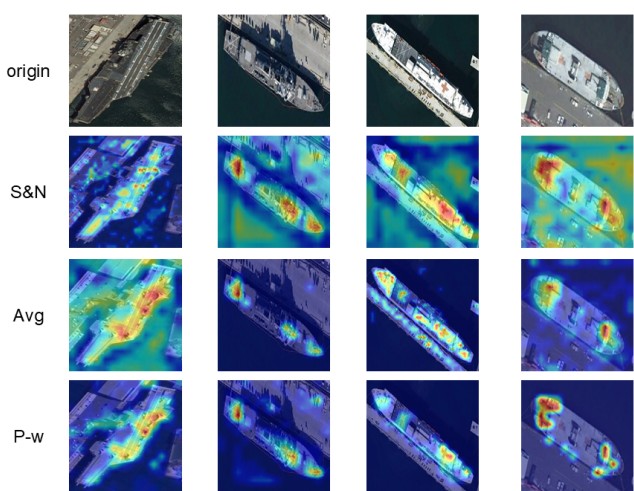

Figure 28: Impact of Hierarchical Prototype

accuracy no longer improves, interpretability weakens. Balancing performance and interpretability, we set $K$=10.

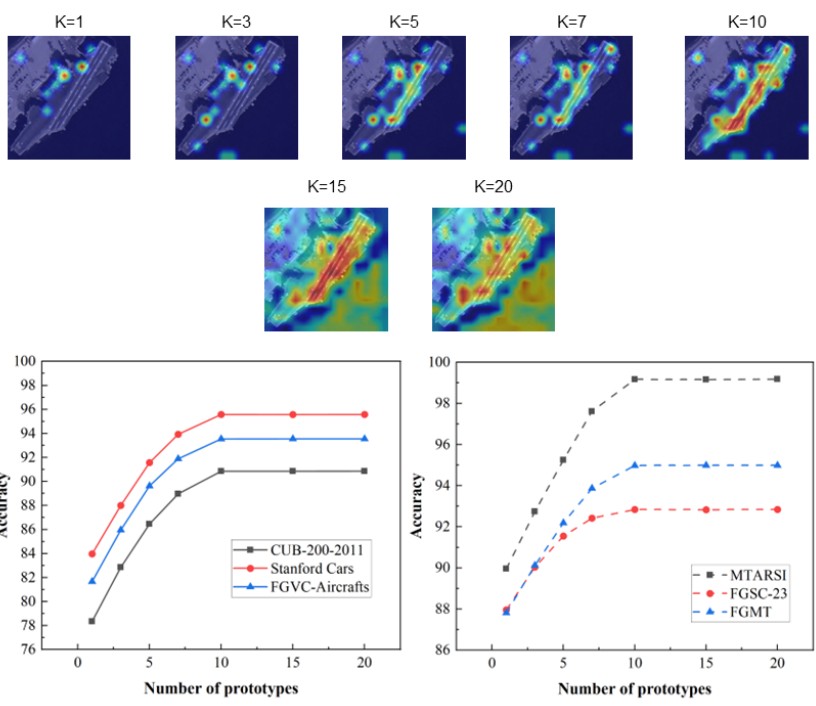

Figure 29: Impact of prototype number K

## A.6 ZERO-SHOT SUPERCLASS GENERALIZATION

Inspired by the Zero-shot Superclass Generalization proposed in the NBDT paper, we construct multiple semantically coherent superclasses across six datasets. Each superclass contains several fine-grained categories. We sample instances from novel categories that never appear during training and input them into the model, then evaluate whether they are correctly mapped to the corresponding

superclass node, thus systematically assessing the model's semantic generalization ability at the superclass level. For "black-box" models, we further examine whether the model can correctly predict a subclass under the correct superclass.

Specifically, we construct superclass hierarchies on CUB-200-2011, Stanford Cars, FGVC-Aircrafts, MTARSI, FGSC-23, and FMGT-28. CUB-200-2011 includes four superclasses: Sparrow (82 samples), Warbler (88 samples), Woodpecker (63 samples), and Gull (51 samples). Stanford Cars also contains four superclasses: Chevrolet (67 samples), Dodge (79 samples), Ford (65 samples), and BMW (62 samples). FGVC-Aircrafts contains three superclasses: Airbus (72 samples), Boeing (58 samples), and McDonnell Douglas (75 samples). MTARSI consists of four superclasses: Strategic Bomber (86 samples), Fighter (64 samples), Transport (58 samples), and Commercial Derivative (48 samples). FGSC-23 is divided into six superclasses: Combat ships (69 samples), Amphibious Ships (71 samples), Support Ships (66 samples), Civilian Ships (58 samples), Strategic Bomber (61 samples), and Fighter (48 samples). FMGT-28 contains six superclasses: Strategic Bomber (61 samples), Fighter (48 samples), Passenger Vehicle (63 samples), Commercial Vehicle (65 samples), Combat ships (58 samples), and Support Ships (47 samples). Table 4 presents the zero-shot superclass generalization results. Bi-HiR exhibits significant superiority in generalization capability compared with their counterpart black-box models and other interpretable models.

### A.7 User Study

#### A.7.1 Questions Setting

This survey on user subjective evaluation of interpretable methods employs a questionnaire format. The survey consists of two distinct questionnaires. Each questionnaire contains 14 questions: 6 questions of the first type and 8 questions of the second type.

For each question, participants will be shown the original image of an aircraft or ship and the explanation result generated by an interpretable method demonstrating the model's prediction process. Considering that participants may lack expertise in the remote sensing domain, we use a Large Language Model (LLM) to generate textual cues describing the target characteristics shown in the original images to aid user responses.During the process of generating text prompts, we strive to adopt objective prompts to minimize their impact on user survey results. We aim to provide reference characteristics for different types of objects solely from the perspective of remote sensing domain knowledge. The prompt we provided to the LLM was: 'You are a remote sensing researcher. I am providing you with an aircraft identification dataset where the target classes are defined as MTARSI_CLASS_NAMES = 0: "A-10", 1: "A-26", 2: "B-1", .... If you were to classify these images, what characteristics would you use as criteria for classifying each category? Please provide these characteristics.' A similar prompt would be used for the ship dataset.

In the first type of question, we display one image correctly predicted by the model and explanation results from different interpretable methods for that prediction. Users will select the explanation result they consider best based on the explanations and by referring to the textual cues.

In the second type of question, we present a series of original images of the same target type along with explanation results from different interpretable methods applied to these images. Participants need to determine whether the model would make a correct prediction based on the explanation results and the textual cues.

To minimize bias arising from differences in how various these interpretable methods generate explanations, the outputs of all methods were uniformly converted into heatmaps, which were then overlaid onto the original images. Additionally, we include a sample image with a color legend at the beginning of the two type of questionnaires to help users understand the feature importance represented by different colors in the heatmaps.

After participants complete all questions, we will collect the questionnaires and organize the response data for subsequent analysis.

#### A.7.2 Examples

As demonstrated in the Type 1 question example (Figure 30), we provide an original aircraft carrier image alongside explanation visualizations generated by multiple interpretable methods interpreting

This question displays an original aircraft carrier image correctly classified by the model and multiple explanation heatmaps generated by different interpretable methods explaining the model's decision. Please analyze these heatmap visualization results and select the option representing the most effective explanation with the help of the following reference features.

**Reference features**: Exceptionally large dimensions, low aspect ratio, prominently large rectangular/polygonal full-length flight deck (the defining characteristic), typically single island superstructure, potential presence of carrier-based aircraft on deck.

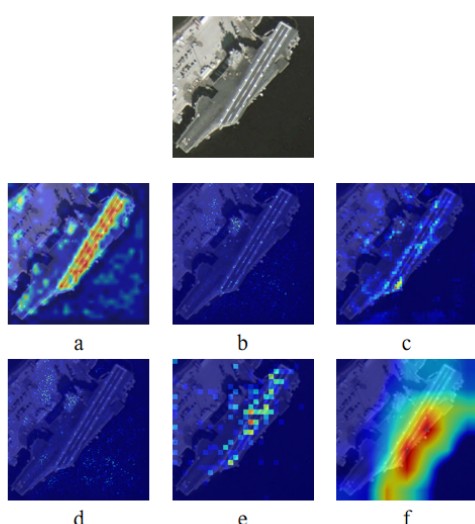

Figure 30: Exampl of questionaire type 1

This question displays a series images of B-1 Lancer bomber where each option features an original image (left) paired with an explanation heatmap (right) generated by a interpretable methods visualizing the model's decision rationale. Please analyze these heatmaps with the help of the following reference features and decide if the model is able to classify these inputs correctly.

**Reference features**: Variable-sweep wing configuration, elongated slender airframe (44-meter length), four underwing-mounted turbofan engines in nacelles, slanted double vertical stabilizers.

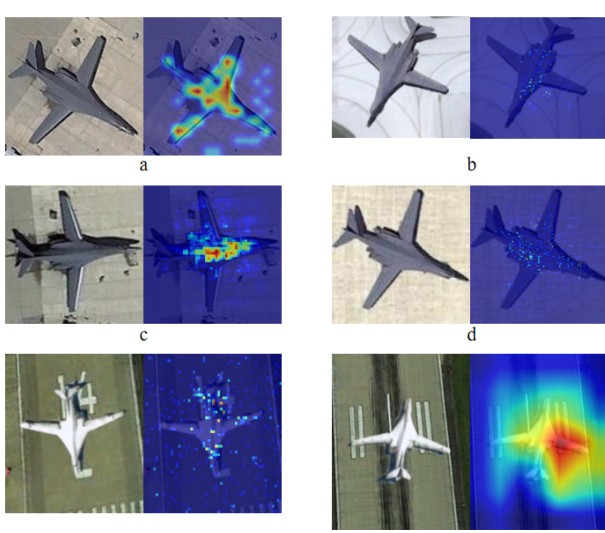

Figure 31: Exampl of questionaire type 1

the model's prediction process; participants then select the most effective interpretable method by evaluating these interpretative outputs and LLM-generated feature reference cues.

Unlike Type 1 questions, Type 2 questions provide explanation results from multiple interpretable methods for a single target class. As illustrated in the B-1 bomber example (Figure 31), we present comparative explanation visualizations across methods; participants then determine which input images the model would predict correctly versus where it would fail, based on these explanations and LLM-generated reference features.

### A.7.3 EVALUATION METRICS

When participants assess whether the model's prediction is correct, their judgments may be either accurate or erroneous. Consequently, their responses can be formalized statistically as a confusion matrix of a binary classifier Yerushalmy (1947), as illustrated in Table 7. True (T) and False (F) denote the model's prediction outcomes, while Positive (P) and Negative (N) represent the correct-

|  | Positive assumption | Negative assumption |
|---|---|---|
| Correct output | True Positive (**TP**) | False Negative (**FN**) |
| False output | False Positive (**FP**) | True Negative (**TN**) |

Table 7: Confusion matrix

ness of the participant's judgment regarding the prediction. This formalization enables quantitative computation of participants' accuracy, sensitivity, and specificity using established classification metrics.

Accuracy quantifies participants' ability to correctly assess model predictions, calculated as the proportion of correct judgments among all responses

$$\text{Accuracy} = \frac{TP + TN}{TP + TN + FP + FN} \quad (8)$$

Sensitivity (True Positive Rate, TPR) measures participants' ability to trust reliable predictions when assisted by post-hoc explanations:

$$\text{TPR} = \frac{TP}{TP + FN} \quad (9)$$

Specificity (True Negative Rate, TNR) evaluates participants' ability to override erroneous predictions using explanations:

$$\text{TNR} = \frac{TN}{TN + FP} \quad (10)$$

In addition, we introduce a metric called *Interpretation Preference*(IP) to quantify user satisfaction with the explanation results generated by interpretable methods:

$$\text{IP} = \frac{x_i}{\sum x_i} \quad (11)$$

where $x_i$ denotes the number of participants who preferred the $i$-th explanation method. This metric measures the proportion of participants who preferred a particular explanation method, thereby reflecting user preference across the various explanation methods.

### A.7.4 QUESTIONNAIRE COLLECTION

We compare our method with current mainstream intrinsically interpretable methods and typical post-hoc explainable methods. Statistical results from 115 users are shown in Table 6 of the main paper. The findings demonstrate that our method significantly outperforms comparative methods in Accuracy: it exceeds the second-highest performing intrinsic method by 25.22% and surpasses the second-best post-hoc method by 19.13%. This indicates that our approach provides substantial advantages in helping users judge model predictions while delivering superior interpretability.

In terms of TPR, our method achieves 21.30% and 16.52% higher scores than intrinsic and post-hoc methods respectively. This demonstrates that when the model makes correct predictions, our explanation algorithm effectively enhances users' trust in the model.

Regarding TNR, our method outperforms intrinsic methods by 25.22% and post-hoc methods by 15.65%. This reveals that when the model generates erroneous predictions, our method helps users question model outputs and understand failure reasons. This capability notably lacking in other interpretable method.

In Efficacy evaluation, our method demonstrates overwhelming superiority: it was selected by 73.62% of users when compared against intrinsic methods and preferred by 79.71% of users versus post-hoc techniques. This signifies that the explanations generated by our method align more closely with human cognitive logic.

### A.8 FAILURE CASE ANALYSIS

Figure 32 shows a misclassification case for Bi-HiR on a single input image, where the model predicts cruiser instead of aircraft carrier. From the visualized reasoning path, the error arises at the

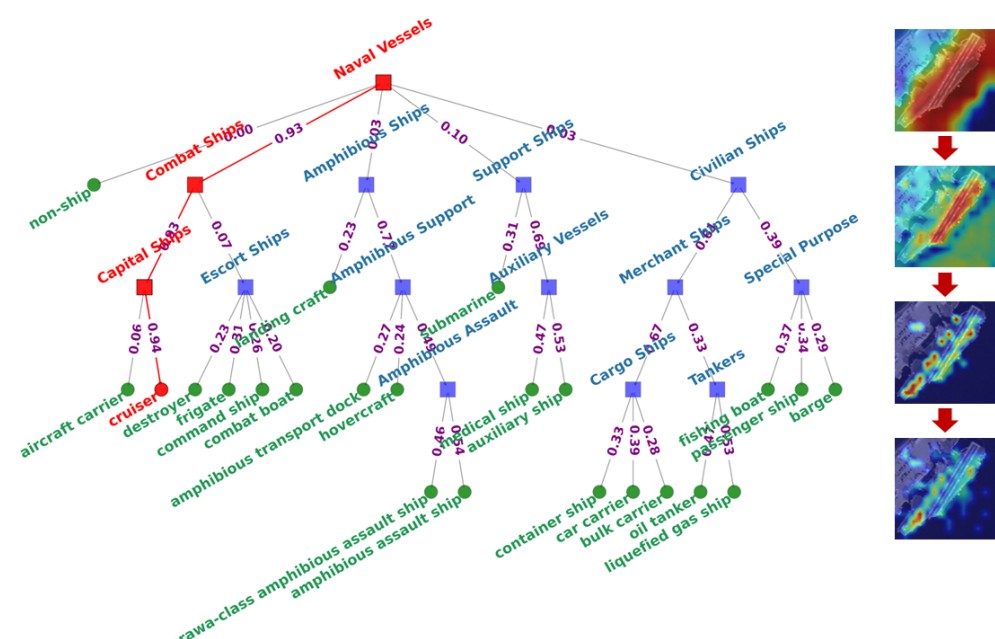

Figure 32: Explanations for Failure Case.

Capital Ships node. The visual explanations along the path indicate that, at the root, the model attends to the entire target, within Combat Ships it concentrates more on the deck, and at Capital Ships it focuses on the deck and bow. However, at the cruiser node, attention shifts to the bow and misses deck cues such as the aircraft runway, causing the misclassification. Although the prediction is incorrect, the node-level trace reveals where the decision failed and demonstrates Bi-HiR's interpretability and error-tracing ability.

## A.9 LIMITATIONS

In this study, we systematically compared the proposed Bi-HiR with several state-of-the-art fine-grained interpretable recognition methods. The results indicate that although Bi-HiR does not achieve the highest classification accuracy, user studies confirm that it provides robust dual interpretability at both the semantic concept level and the visual level. By sacrificing a small amount of recognition accuracy, the model attains stronger interpretability, a trade-off that is reasonable and acceptable in scenarios emphasizing transparency and auditability. Future work will focus on further improving recognition performance while continuing to enhance interpretability, aiming for a synergistic advancement of both aspects.

## A.10 THE USE OF LARGE LANGUAGE MODELS

This paper used a large language model (OpenAI ChatGPT) in two ways: (i) to polish the English writing for clarity and fluency, and (ii) as an auxiliary tool—via the GPT API—to construct the semantic hierarchy in our interpretability pipeline. The model did not generate research ideas, experiments, or conclusions; all ideas, methodologies, and final decisions are solely those of the authors.

