# OpenReview forum: "Bidirectional Hierarchical Reasoning for Fine-grained Visual Recognition"
_ICLR.cc/2026/Conference — Submitted to ICLR 2026_

### Official Review · Reviewer_kvuV · 2025-10-17

**Soundness:** 1
**Presentation:** 2
**Contribution:** 2
**Rating:** 2
**Confidence:** 5

**Summary:**

This work studies the task of Fine-grained visual recognition (FGVR), which demands not only high classification accuracy but also human-aligned interpretability that captures the natural coarse-to-fine reasoning hierarchy of human cognition. To meet this need, the work proposes Bi-HiR (Bidirectional Hierarchical Reasoning), a  framework that emulates hierarhical probabilities by integrating top-down semantic guidance with bottom-up prototype-based explanations. Bi-HiR automatically constructs the coarse-to-fine semantic hierarchy using Large Language Model (LLM)-derived semantic priors, eliminating the need for costly manual annotations, and utilizes a joint optimization strategy where these priors guide the learning of hierarchical prototypes. Experiments were conducted on six FGVR benchmarks (CUB-200, Cars, Aircraft, MTARSI, FGSC-23, and FMGT-28). Lastly, the evaluations demonstrated that Bi-HiR exhibits superior zero-shot superclass generalization and its interpretability results in higher human trust, as confirmed by a user study.

**Strengths:**

- The paper is well-written and easy to understand.
- The idea of using hierarchical decision process to make FGVR more interpretable is practical.
- The enhanced human trust and error diagnosis achieved by Bi-HiR is impressive. This shows its advantage on the interpretability aspect for FGVR.

**Weaknesses:**

### **Primary Weakness**

**(1) Lack of proper citation to original technical innovations**

The “proposed” idea and method of using LLMs to automatically obtain coarse-to-fine categorical hierarchies and visual attributes are **not new** and do not originate from this work.

Regarding hierarchical label acquisition with LLMs, this idea and technique were first proposed and utilized in **CHILS** [1] and **SHINE** [2], where GPT-3.5 was employed to derive three-level label hierarchies that improved the zero-shot open-vocabulary recognition performance of VLMs and detectors.

Regarding visual attribute acquisition with LLMs and subsequent training, these ideas and techniques were: (1) originally proposed by **FineR** [3]; and (2) later extended in **FineDefics** [4], which used such pseudo-labeled attributes as auxiliary supervision to enhance the fine-grained recognition performance of large VLMs.

Nevertheless, the authors have neither discussed nor cited **CHILS**, **SHINE**, **FineR**, and **FineDefics**, nor have they acknowledged these prior works when describing the innovation. **The authors should appropriately credit previous research contributions rather than implying that these ideas originate solely from this work.**

---

**(2) Lack of novelty compared to prior works**

A follow-up concern is that, given these core techniques and ideas were already proposed in prior works as described above, what is the actual technical or conceptual novelty of this work—beyond adapting and combining existing methods for the hierarchical recognition task?

Furthermore, regarding the loss function design, what is the difference between the proposed regularization terms and the techniques used in prior work [5] (particularly Sections 3.1–3.2 in that paper)?

---

**(3) Why is Bi-HiR better?**

From the experimental results, the proposed **Bi-HiR** model actually **consistently underperforms** compared to the state-of-the-art method (**CGL**) when using the same ResNet-50 backbone. Therefore, could the authors qualitatively justify the potential value or advantages of Bi-HiR? To clarify, it is acceptable for Bi-HiR to underperform SOTA models, as long as it provides distinct benefits or insights of its own.

---

**References**

[1] Novack, Z., McAuley, J., Lipton, Z. C., & Garg, S. (2023, July). Chils: Zero-shot image classification with hierarchical label sets. In ICML, 2023.

[2] Liu, M., Hayes, T. L., Ricci, E., Csurka, G., & Volpi, R. (2024). Shine: Semantic hierarchy nexus for open-vocabulary object detection. In CVPR, 2024.

[3] Liu, M., Roy, S., Li, W., Zhong, Z., Sebe, N., & Ricci, E. (2024). Democratizing fine-grained visual recognition with large language models. In ICLR, 2024.

[4] He, H., Li, G., Geng, Z., Xu, J., & Peng, Y. (2025). Analyzing and boosting the power of fine-grained visual recognition for multi-modal large language models. In ICLR 2025.

[5] Bertinetto, L., Mueller, R., Tertikas, K., Samangooei, S., & Lord, N. A. (2020). Making better mistakes: Leveraging class hierarchies with deep networks. In *CVPR, 2020.*

**Questions:**

- The github link provided in the Abstract for code cannot be reached (404 error).
- The presentation of this manuscript is too dense. Without reading the full text, it makes the reader hard to understand Fig 1 and Fig 2. They are overwhelming. Fig 2 contains too many architectural details. The reviewer suggests to remove the details to make it only illustrate the abstract level pipeline.
- The reviewer is confused by the “reasoning” process claimed in this work. The decision is made in a hierarchical manner based on predicted probability. What’s is the reasoning here?
- Is there any point in including the published venue on comparative methods? The reviewer does not get any useful information from this.
- Please make the comparison under the same model size and training data. Otherwise, it is hard to compare.

---

### Official Review · Reviewer_9w2r · 2025-10-29

**Soundness:** 2
**Presentation:** 2
**Contribution:** 2
**Rating:** 2
**Confidence:** 5

**Summary:**

The paper (1) uses LLM to construct coarse-to-fine hierarchies without manual annotations; (2) introduce a joint optimization strategy, and (3) produces interpretable, step-wise visual and semantic explanations.

**Strengths:**

Pros:
1. The paper considers an interesting problem,
2. The writing is easy to follow.

**Weaknesses:**

1. The code of this paper is changed after the submission deadline. https://github.com/dudududa-max/Bi-HiR It is not allowed in ICLR.
2. The paper’s core ideas shares similarity with TransHP [1], which can also be understood as coarse-to-fine reasoning/prompting process. It is not cited.
3. As shown in Fig. 2, the paper is much like a combination of the existing techniques. Therefore, I doubt the novelty of the paper.
4. Although the proposed method is complicated, it stills does not achieve SOTA as shown in Table 1. The effectiveness of this method is not verified.

**Questions:**

NA

---

### Official Review · Reviewer_ZkNv · 2025-11-01

**Soundness:** 3
**Presentation:** 3
**Contribution:** 3
**Rating:** 4
**Confidence:** 3

**Summary:**

This paper introduces Bi-HiR, a Bidirectional Hierarchical Reasoning framework that emulates human-like coarse-to-fine cognition for fine-grained visual recognition (FGVR). Bi-HiR integrates top-down semantic reasoning, derived automatically from LLM-generated semantic hierarchies, with bottom-up prototype learning for interpretable and accurate recognition. The hierarchy is constructed through GPT-4o prompts (no manual annotation) and used to guide prototype optimization via bidirectional semantic–visual alignment.
Experiments on six FGVR benchmarks, CUB-200, Cars, Aircraft, MTARSI, FGSC-23, and FGMT-28, show that Bi-HiR achieves:
- Top-3 accuracy among both interpretable and black-box baselines;
- Improved zero-shot superclass generalization, demonstrating semantic alignment;
- Strong human-trust results in a 115-participant interpretability study.
Overall, Bi-HiR presents a unified interpretable reasoning model that blends concept bottleneck (CBM) and prototype (PPN) paradigms through a coherent bidirectional optimization.

**Strengths:**

1. The bidirectional reasoning design elegantly bridges top-down and bottom-up interpretability. The model emulates human hierarchical cognition while maintaining strong predictive accuracy.
2. Employing GPT-4o to construct coarse-to-fine semantic structures is a practical and scalable innovation, eliminating manual ontology engineering and allowing domain adaptation.
3.  Results span six datasets, extensive ablations (modules a–e, LT-sep analysis), zero-shot superclass generalization, and a well-designed human-trust user study.
4. The inclusion of subjective interpretability metrics (Accuracy, TPR, TNR, IP) provides quantitative evidence for improved user trust, rare among FGVR works.
5. The paper includes detailed architecture, prompt examples, loss functions (Eqs. 1–12), and compliance statements on ethics and reproducibility.

**Weaknesses:**

1.  While the optimization (Eqs. 1–12) is well-defined, the paper lacks theoretical analysis on convergence, semantic consistency, or the interpretability–accuracy trade-off.
2. The reliance on GPT-4o limits reproducibility and raises fairness concerns. Testing hierarchy generation with open models (e.g., LLaMA-3, Mistral-Large) would strengthen robustness claims.
3. The “human-like reasoning” claim remains confined to visual FGVR. Demonstrating transfer to multimodal or text–vision reasoning tasks could broaden impact.
4. Although the 115-participant study is commendable, methodological details (randomization, significance tests, inter-rater reliability) are insufficient. Reporting confidence intervals or effect sizes would reinforce validity.
5. Missing Contemporary References (2025) – The discussion omits several highly relevant 2025 works that would strengthen contextualization within the current landscape of interpretable reasoning:
- IVPT (Wang et al., 2025) – Exploring Interpretability for Visual Prompt Tuning with Hierarchical Concepts; introduces hierarchical concept alignment for interpretable visual prompt tuning.
- HCG-LVLM (Guo et al., 2025) – Hierarchical Contextual Grounding LVLM: Enhancing Fine-Grained Visual-Language Understanding with Robust Grounding; advances hierarchical reasoning for multimodal alignment.
- Tree-based VLM Reasoning (Elmansoury et al., 2025) – Decomposing Visual Classification: Assessing Tree-Based Reasoning in VLMs; critically analyzes the limits of explicit hierarchical reasoning in large vision-language models.
- Causal-FGVC (Zhang et al., 2025) – Learning High-Order Features for Fine-Grained Visual Categorization with Causal Inference; introduces a causal-inference-based framework for generalizable fine-grained recognition.
Incorporating discussion of these studies would better situate Bi-HiR within the 2025 research frontier on hierarchical, causal, and interpretable reasoning in vision systems.

**Questions:**

How sensitive is the automatically generated hierarchy to prompt variations or stochastic LLM outputs?
Could open-source models (e.g., LLaMA-3, Mistral-Large) approximate GPT-4o’s hierarchy generation quality?
Does the LT-sep loss remain stable for deeper hierarchies (> 4 levels), or does it risk gradient dilution?
How would Bi-HiR perform on multimodal (image–text) datasets to substantiate its “cognitive reasoning” claim?
Could uncertainty propagation (e.g., entropy-based routing confidence) be incorporated for calibrated explanations?

---

### Official Review · Reviewer_caeo · 2025-11-02

**Soundness:** 2
**Presentation:** 2
**Contribution:** 2
**Rating:** 2
**Confidence:** 4

**Summary:**

The paper proposes Bi-HiR, a bidirectional hierarchical reasoning framework for fine-grained visual recognition that integrates top-down semantic priors from large language models (LLMs) with bottom-up prototype-based explanations. The goal is to improve interpretability and human trust while maintaining competitive recognition accuracy. Experiments are conducted on multiple fine-grained datasets, and the authors claim that Bi-HiR achieves both strong performance and superior interpretability.

**Strengths:**

1. interesting integration of LLM priors
The paper’s idea of using large language models to automatically construct semantic hierarchies is interesting, offering a scalable alternative to human-annotated concept hierarchies.
2. Reproducibility
The code is publicly accessible.

**Weaknesses:**

1. Novelty is limited.
The proposed framework appears to be a combination of existing techniques (Grad-CAM enhancement, multi-scale fusion, prototype-based reasoning) rather than a fundamentally new algorithmic contribution. Each component has been explored before, and the integration does not seem to introduce a novel theoretical insight or a clearly justified synergy among modules.
2. Unclear evaluation of “better explanations”
The paper claims improved interpretability, but this is hard to measure or verify. For example, in Figure 3, the proposed method shows smaller highlighted regions — but it is unclear whether a smaller region necessarily indicates a better explanation. The visual evidence is subjective and lacks quantitative metrics such as localization accuracy,  or concept alignment.

3. Unfair or unclear comparisons
Figure 3 only shows visualizations for the 2×2 grid inputs for Bi-HiR, while other methods’ single-input attention maps are missing. To ensure fairness, attention maps for both single and composite inputs from all baselines should be compared.
Moreover, the meaning and motivation of using a “2×2 grid input” are unclear and seem artificial and not meaningful for evaluating fine-grained recognition.

4. Ambiguous reporting in Table 1
The paper reports “Top-3” rankings for Bi-HiR, which may overstate performance. It would be more informative to show full rankings among interpretable methods only, or at least ensure all methods are compared under the same backbone (e.g., ResNet-50 or ConvNeXt-B). Without consistent backbones, the performance comparison is not convincing. It's better to show performance by group, i.e. same backbone group; interpretable method group.

5. Overly complex and unclear framework design
Figure 2 is visually overloaded and difficult to follow. The overall system looks like a collection of loosely connected modules rather than a coherent, streamlined framework. The flow between “Post-hoc Enhancement,” “Multi-scale Aggregation,” and “Bi-HiR Training” stages is not clearly explained.

6. Contradictory ablation results (Table 2)
According to Table 2, removing components (c) and (d) (semantic hierarchy and LT-sep loss) actually improves performance, raising questions about their necessity. If these components degrade accuracy, what exactly is gained in terms of interpretability? Is the explanation significantly better with them included? This trade-off should be clearly quantified and justified.

**Questions:**

see above.

---

### Meta-Review · Area_Chair_bWXP · 2025-12-15

**Summary:**

After reading the manuscript and reviewers' comment (the authors did not provide their response), I made my recommendation *reject*. Here are the detailed meta review.

**Research Question**

The authors consider the well-defined visual cognition problem.

**Motivation**

The authors argue that the existing methods lack hierarchical interpretation, which aligns with human cognitive processes.

**Philosophy**

The authors aim to consider such hierarchy in visual recognition.

**Solution**

The authors propose Bidirectional Hierarchical Reasoning (Bi-HiR) framework by integrating both top-down and bottom-up reasoning.

**Experiments**

The authors conduct extensive experiments on visual recognition. However, it lacks verification on the interpretation. Figure 3 is not enough to demonstrate the interpretation of the proposed method aligns with human cognition. This point makes this submission not self-standing.

**Presentation**

1. Some citation formats are incorrect.

2. It is not helpful to place an experimental figure on the first page, which contains too much information. Readers feel difficult to understand before reading the proposed method. Usually, a motivation figure would be helpful.

3. A punctuation mark is needed at the end of each equation.

**Summary**

I have to agree with the reviewers' comments. (1) The proposed framework appears to be a combination of existing techniques (Grad-CAM enhancement, multi-scale fusion, prototype-based reasoning) rather than a fundamentally new algorithmic contribution. Each component has been explored before, and the integration does not seem to introduce a novel theoretical insight or a clearly justified synergy among modules. (2) The paper claims improved interpretability, but this is hard to measure or verify. (3) Although the provided github link is invalid, it is also risky to exposure the author identity. (4) There is no discussion of the trade off between recognition performance and interpretation.

**Reviewer Concerns:**

Since there is no author response, the original concerns exist.

**Reviewer Scores:**

Since there is no author response, reviewers would not change their scores.

---

### Decision · Program_Chairs · 2026-01-26

Reject